# ActiView: Evaluating Active Perception Ability for Multimodal Large Language Models

## Abstract

Active perception, a crucial human capability, involves setting a goal based on the current understanding of the environment and performing actions to achieve that goal. Despite significant efforts in evaluating Multimodal Large Language Models (MLLMs), active perception has been largely overlooked. To address this gap, we propose a novel benchmark named ActiView to evaluate active perception in MLLMs. Since comprehensively assessing active perception is challenging, we focus on a specialized form of Visual Question Answering (VQA) that eases the evaluation yet challenging for existing MLLMs. Given an image, we restrict the perceptual field of a model, requiring it to actively zoom or shift its perceptual field based on reasoning to answer the question successfully. We conduct extensive evaluation over 27 models, including proprietary and open-source models, and observe that the ability to read and comprehend multiple images simultaneously plays a significant role in enabling active perception. Results reveal a significant gap in the active perception capability of MLLMs, indicating that this area deserves more attention. We hope that our benchmark could help develop methods for MLLMs to understand multimodal inputs in more natural and holistic ways.

## 1 Introduction

The advent of Multimodal Large Language Models (MLLMs) has marked a significant milestone in the realm of artificial intelligence, demonstrating capabilities that are increasingly approaching human-like performance (OpenAI, 2023; Liu et al., 2023c; Ye et al., 2024b). This advancement, while promising, also presents new challenges and opportunities for evaluating these models. As a result, the landscape of MLLM evaluation is rapidly evolving, with numerous benchmarks being developed to either comprehensively evaluate models (Fu et al., 2023; Liu et al., 2023d; Fu et al., 2024a) or to analyze specific aspects of their capabilities (Liu et al., 2023a; Lu et al., 2023; Luo et al., 2024; Xiao et al., 2024; Li et al., 2024b; Nie et al., 2024; Qian et al., 2024).

Despite the extensive efforts devoted to MLLM evaluation, *active perception* (Bajcsy, 1988; Bajcsy et al., 2018) remains underexplored. Active perception involves understanding the reasons for sensing, choosing what to perceive, and determining the methods, timing, and locations for achieving that perception (Bajcsy et al., 2018). This is important because in the real world, the desired information often does not appear directly in the center of one's field of vision. Instead, it requires individuals to move their field of view, locate details, and filter out distracting information. For example, in Figure 1a, suppose we are looking for information in a giant painting. We need to first shift our view to locate the specific area and then possibly zoom in to gather detailed information. Intuitively, active perception not only enables a person or model to accomplish more complex tasks, but it also has the potential to serve as a good indicator of the level of intelligence of a model. This makes it a critical capability that warrants thorough evaluation.

However, existing multimodal evaluation benchmarks are not well-suited for measuring active perception capabilities. We summarize several widely used or recently proposed multimodal evaluation benchmarks in Table 1. Most of these benchmarks assess models in static perceptual field settings, evaluating how well a model can process information presented directly to it without requiring active exploration or dynamic adjustments to its field of view. BLINK (Fu et al., 2024b), V* (Wu & Xie, 2023), and CNT (Roberts et al., 2023) are the only exceptions, as they utilize dynamic perceptual fields. However, they only consider either shifting or zooming of the field of view in specific

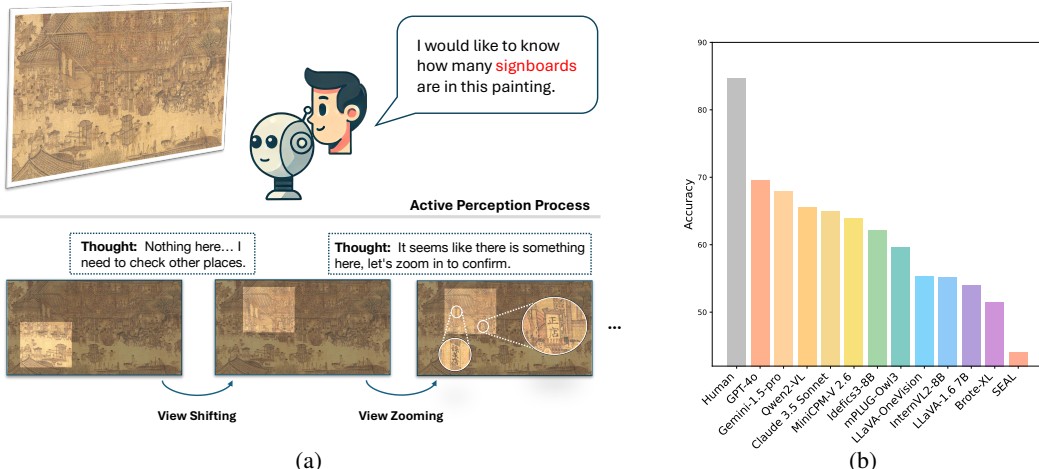

Figure 1: (a) Active perception allows humans or models to perform more complex tasks by actively seeking and processing relevant information. In this paper, we evaluate two key active perception abilities for MLLMs: 1) *shifting*, as real-world scenarios often present limited views and require shifts to obtain new perspectives, and 2) *zooming*, which helps enhance perception by zooming out for broader views and zooming in for details. (b) Model results on our benchmark compared to human performance, showing that our benchmark is challenging for models.

scenarios, which are insufficient for measuring active perception capabilities. Therefore, there is a clear need for new evaluation frameworks that can adequately assess active perception abilities across diverse and dynamic environments.

To fill this gap, we introduce a novel benchmark specifically designed for evaluating **Acti**ve perception through **View** changes (**ActiView**). Given the difficulty of comprehensively evaluating active perception capabilities across all possible scenarios, ActiView concentrates on a series of tasks that are currently feasible to evaluate yet still present significant challenges to current models. The evaluation instances in ActiView follow the Visual Question Answering (VQA) (Antol et al., 2015) format but include the following additional features: 1) Each question requires an understanding of detailed visual clues in the image to answer accurately. 2) We impose view constraints on images, allowing the model to perceive only a partial field of view of the full image at a time. This setup explicitly requires models to perform view shifting and zooming to gather more information and eliminate potential distractions, thereby simulating the active perception process. We collect images from existing datasets as well as newly captured images, carefully curate questions, answers, and visual clues manually to ensure the quality and diversity of the benchmark.

We conducted evaluations on three proprietary models and 24 widely used and advanced open-source models. The results reveal that these models generally lag behind in active perception. For instance, the strong proprietary model, GPT-4o, only achieved an average score of 66.40% with our designed evaluation pipelines, and a score of 67.38% on the general VQA format, which is significantly lower than the human performance score of 84.67%. Moreover, the average performance gap between proprietary models and open-source models in active perception is considerably smaller within our designed evaluation pipelines compared to gaps observed in other tasks from previous research. Experimental results suggest that models tend to perform better when the full image is provided, while fall shot to form a holistic perspective of the complete image when they only have access to all the constrained perceptual fields. Therefore, we believe that active perception warrants more research efforts, and our benchmark is valuable for further study.

## 2 RELATED WORK

### 2.1 MLLM BENCHMARKS

Extensive efforts have been devoted to developing MLLM evaluation benchmarks (Table 1). These benchmarks cover a wide range of capabilities, including but not limited to visual comprehension (Fu et al., 2023; Liu et al., 2023d; Fu et al., 2024a), visual perception (Fu et al., 2024b), hallucination (Liu et al., 2023a), and mathematical and logical reasoning (Lu et al., 2023; Xiao et al., 2024). However, most of these benchmarks utilize a static view of the input image, which is

Table 1: Comparison with other benchmarks for MLLMs. "Per. Fields": Perceptual Fields. 1.9k*: Videos. Manual*: A mixture of manual annotation and data from existing benchmarks. Our benchmark concentrates on evaluating active perception abilities via the change of visual perceptual fields, including shifting to different fields for compensating missing information, and zooming for fine-grained details in the current fields.

| Benchmarks | Evaluation Target | Change of Per. Fields | | Num. | Evaluation | Annotator |
| | | Shifting | Zooming | Img | Instances | |
|---|---|---|---|---|---|---|
| MME (Fu et al., 2023) | Visual comprehension | ✗ | ✗ | 1.1k | 1.3k | Manual |
| MMBench (Liu et al., 2023d) | Visual comprehension | ✗ | ✗ | 1.8k | 1.8k | Manual + Auto |
| MM-Vet (Yu et al., 2023) | Integrated capabilities | ✗ | ✗ | 200 | 218 | Manual* |
| Seed-Bench (Li et al., 2023b) | Visual comprehension | ✗ | ✗ | 1.9k* | 24k | Auto |
| BLINK (Fu et al., 2024b) | Visual perception | ✓ | ✗ | 7.3k | 3.8k | Manual |
| Video-MME (Fu et al., 2024a) | Visual comprehension | ✗ | ✗ | 0.9k | 2.7k | Manual |
| ViP-Bench (Biernacki et al., 2021) | Understanding of visual prompt | ✗ | ✗ | 303 | 303 | Manual |
| V* (Wu & Xie, 2023) | Visual search for fine-grained details | ✗ | ✓ | 191 | 191 | Manual |
| HallusionBench (Liu et al., 2023a) | Hallucination | ✗ | ✗ | 346 | 1.1k | Manual + Auto |
| CNT (Roberts et al., 2023) | Geographic and Geospatial | ✗ | ✓ | 345 | 345 | Manual |
| BenchLMM (Cai et al., 2023) | Cross-style visual comprehension | ✗ | ✗ | 1.7k | 1.7k | Manual + Auto |
| CODIS (Luo et al., 2024) | Context-dependent visual comprehension | ✗ | ✗ | 377 | 754 | Manual |
| LogicVista (Xiao et al., 2024) | Visual logical reasoning | ✗ | ✗ | 448 | 448 | Manual |
| ActiView (Ours) | Active perception | ✓ | ✓ | 314 | 1,625 | Manual |

not suitable for evaluating active perception. While BLINK (Fu et al., 2024b) involves view shifting and both V* (Wu & Xie, 2023) and CNT (Roberts et al., 2023) require view zooming, active perception is not a prerequisite for solving their evaluation questions, making them insufficient for active perception evaluation. In contrast, our benchmark considers both view shifting and zooming. Questions in our benchmark are specifically designed to necessitate active perception for answering, making it a more comprehensive benchmark for active perception evaluation. Active perception is also used to study the control of model states involving dynamic interactions between models and environment (Oh et al., 2016; Ammirato et al., 2017). However, in this paper, we focus on how models actively perceive and understands visual information, which are orthogonal to these works.

## 2.2 ACTIVE PERCEPTION IN MLLMs

Although MLLMs have attracted extensive interest, less effort has been dedicated to improving the active perception capability of MLLMs. One line of research focuses on improving the ability of processing high-resolution images by using higher-resolution ViTs (Ye et al., 2024b), slicing high-resolution images and then concatenate them (Liu et al., 2024), or directly using LLMs to process raw patches of any resolution (Li et al., 2023a). The other line emphasizes visual search for fine-grained details. For example, SEAL (Wu & Xie, 2023) fine-tunes a framework of two MLLMs to follow the visual search mechanism for precise visual grounding, and V-IRL (Yang et al., 2024) proposes an active detection strategy to improve the comprehension of real-world geospatial information. Despite these efforts, our evaluation results reveal that existing MLLMs still generally lack active perception capabilities. Our benchmark will shed light on evaluating and enhancing active perception in MLLMs.

## 3 ACTIVIEW

Our benchmark exams active perception abilities of models via different perceptual fields, where **Acti**vely zooming and shifting of **View**s (**ActiView**) are required. We summarise zooming and shifting as two fundamental factors of active perception as illustrated in Figure 2, where we can evaluate active perception abilities of models through the two factors separately or integratedly. ActiView imitates the procedure of active perception by providing models with an initial view, which is a cropped field of the original image, as depicted in Figure 2, and requires models to search for missing but important information with view zooming and shifting, and to minimize distractions caused by redundant information in the view. We manually annotated 325 questions, where each question corresponds to 5 different evaluation instances that assess active perception abilities under different settings and levels of difficulty. In total, there are 1,625 evaluation instances in our benchmark. This section elaborately describes ActiView benchmark, including the design of categories, questions and options (§3.1) and the data collection procedure and statistics of the benchmark (§3.2).

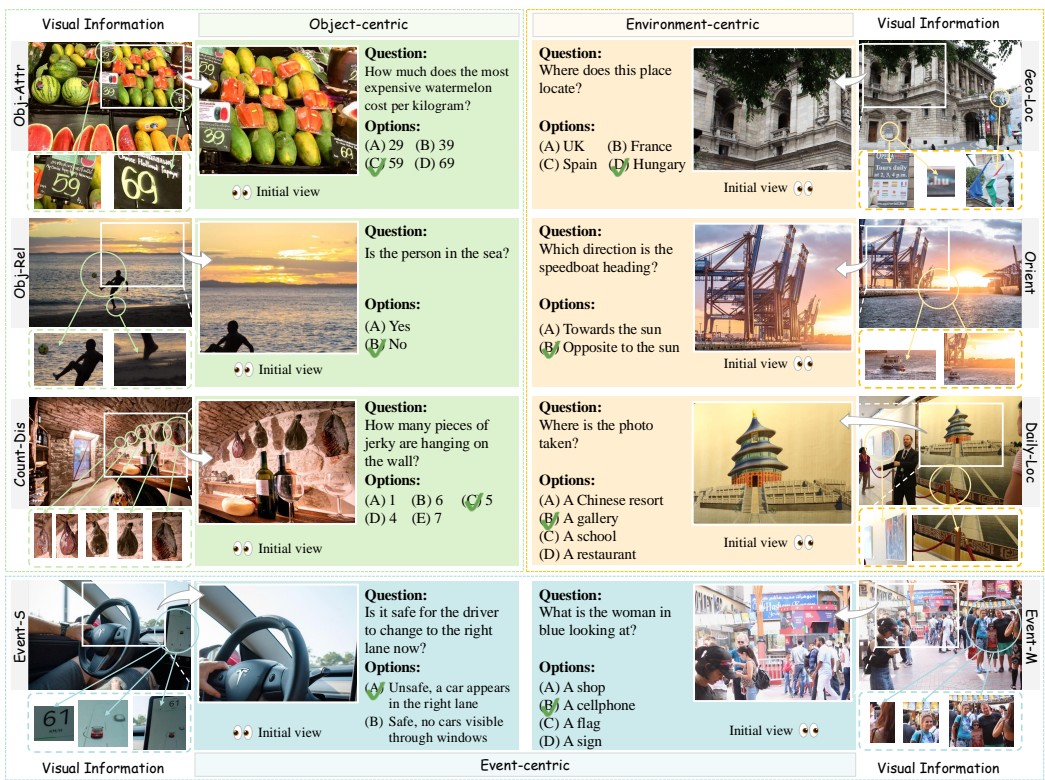

Figure 2: Examples of ActiView. Instances in ActiView present the following features: i) requiring focusing on multiple fine-grained regions; ii) requiring distinguishing distracting information from the entire image; iii) requiring moving of perceptual fields to obtain sufficient visual information to answer questions. During evaluation, models will be given an initial view cropped from the original image as shown above. The evaluation procedures given the initial views are introduced in §4.

## 3.1 BENCHMARK OVERVIEW

When perceiving an image, humans intuitively focus on three principle aspects: the environment depicted in the image, the primary objects, and the event that these objects are engaged in. Correspondingly, we summarise the questions in our benchmark into three main categorises, including environment-centric (Type I), object-centric (Type II), and event-centric (Type III) categorises. These main categorises are further divided into eight sub-classes according to the specific type of visual information and visual features used for answering the questions. Figure 2 displays eight examples for each of the sub-classes, and demonstrates features of our benchmark. For the environment-centric category, three sub-classes are developed:

- **Geo-Localization**, denoted as *Geo-Loc*. This sub-class focuses on geographical features that are unique to a country or a city, and requires models to identify geographical locations depicted in target images. Typical questions for this sub-class are "*Where is this place located?*","*In which country is the photo taken?*", and etc. Images in this sub-class usually contain unique landmarks such as the Eiffel Tower in Paris and the Atomium in Brussels.
- **Orientation**, denoted as *Orient*. This sub-class challenges models to exploit natural orientation information for answering the questions, such as the position of shadows, the position of the sun, and the directional information on street signs. Questions of this type include "*Is this a sunset or a sunrise?*", "*Where is the sunlight coming from?*" and etc.
- **Daily-location**, denoted as *Daily-Loc*. To distinguish from the Geo-Loc sub-class, this subclass concentrates on locations in everyday life that could appear in most of the cities and are not unique to a certain city or country. Images in this sub-class usually depict scenes of museums, restaurants, shops, etc. The corresponding questions include "*Where is this picture most likely taken?*", "*Is there a music school nearby?*", and etc.

For the object-centric category, we expect models to exhibit abilities beyond simple grounding tasks that directly ask for the attributes or relations of objects. Questions for this category usually involve

distracting information from images, and require models to precisely understand the intentions. Sub-classes are demonstrated as follows:

- **Object-attribute**, denoted as *Obj-Attr*. These questions ask for the attributes of objects, with the appearance of distracting information that potentially lead to incorrect answers. As the example of Obj-Attr shown in Figure 2, the highest price, 69 per kilogram, corresponds to papaya rather than watermelon.
- **Object-relation**, denoted as *Obj-Rel*, which concentrates on the spatial relationships among multiple objects. Questions in this sub-class do not directly ask for the spatial relationships. They require models to reason for the correct answer via spatial information. Figure 2 displays an example of Obj-Rel in which models should be aware of the relative positions of the feet of the person to the water.
- **Counting**, denoted as *Count-Dis*. Although this sub-class focuses on the number of objects, different from the counting tasks in other benchmarks (Fu et al., 2023; Yu et al., 2023), there are similar but distracting information about the targets in the images. These distracting objects easily confuse models and challenge the abilities to understand and strictly follow instructions. As the Count-Dis example shown in Figure 2, the jerky on the table should not be considered when answering the question of "*How many pieces of jerky are hanging on the wall?*".

The event-centric category focuses on the interactions of humans and items, such as movements, actions and activities. This category is divided according to the number of objects involved in the target event as following:

- **Event-single**, denoted as *Event-S*. There is only one item or person involved in the target event. For example, the image for Event-S in Figure 2 shows one person driving without other people presenting in the image.
- **Event-multi**, denoted as *Event-M*. Different from Event-S, events of this type happen among multiple items or people. As the example of Event-M shown in Figure 2, the "*woman in blue*" is engaged in a photo shooting activity in which she is posing and another person is taking photo for her. This type requires models to distinguish the event (or events) each entities presenting in the image are engaged in.

To prevent MLLMs from directly answering the questions by merely going through the options, we formulate most of candidate options from the images themselves. For instance, as the example of Geo-Loc shown in Figure 2, there are corresponding information for each of the options in the image, where flags representing UK, France, Spain and Hungary appear. Additionally, for options comprised by numbers, we display them in random order to avoid biased predictions.

## 3.2 DATA COLLECTION AND DATA STATISTICS

Our data are manually curated, including image collection and question annotation processes. To evaluate active perception abilities requiring zooming of fine-grained details and shifting views to obtain missing information, we carefully select images containing multiple fine-grained objects and depicting complex environment and events. To assure the clearness of visual details and the quality of initial views, we collect high-resolution images as described in Appendix A, together with annotation guidelines to ensure the fairness and reliability of question-option-answer triplets. Additionally, annotators are instructed to identify and

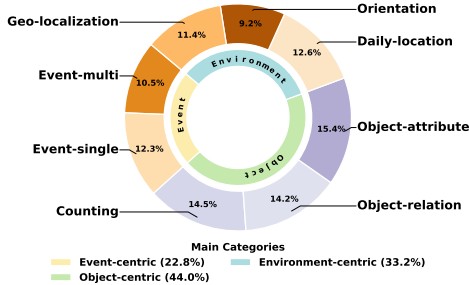

Figure 3: The distribution of categories and sub-classes of our benchmark.

highlight the corresponding visual clues in the image that support their answer. In Figure 2, the human-annotated visual clues are presented as sub-images placed below the original images in "Visual Information" columns. Furthermore, these visual clues will serve as criteria for automatic sorting of different difficulties. We also discuss approaches for automatic data generation in Appendix K. As a brief conclusion here, powerful models, such as GPT-4V , fail to satisfy our annotation criteria, because they suffer from hallucination when comprehending images, and fail to distinguish visual facts from the image and external world knowledge that does not present in the image.

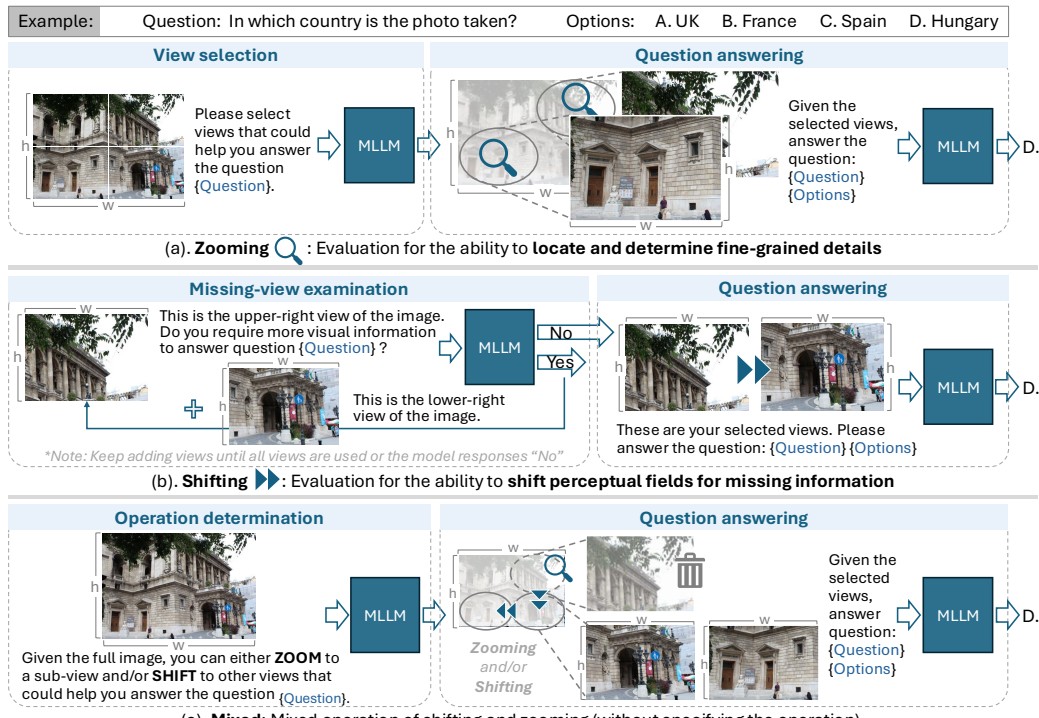

Figure 4: Evaluation pipelines for active perception abilities as described in §4.2.

The distribution of categories are shown in Figure 3. We manually collected 314 images and annotated 325 questions. For each of the questions, the number of candidate options ranges from two to seven, with an average of 3.24 options per question. There are 2.64 sub-views containing human-annotated clues for each of the questions on average. This indicates that a single view is not sufficient for obtaining the correct answer, and that the ability to read and comprehend multiple images is indispensable to our benchmark.

## 4 EVALUATION

To conduct thorough investigation of active perception abilities of MLLMs, we design three evaluation pipelines for different operations of perceptual fields as illustrated in Figure 4, including two fundamental pipelines that apply zooming and shifting respectively, and a mixed pipeline comprising both zooming and shifting operations. We set up five different initial views for each question-image pairs, where a full initial view is used for the zooming and the mixed pipeline, and four constrained views are applied for the shifting pipeline.

### 4.1 MODELS

We investigate both proprietary and open-source models. The proprietary models include widely discussed GPT-4o (OpenAI, 2024), Gemini-1.5-pro (Reid et al., 2024), and Claude 3.5 Sonnet (Anthropic, 2024). For open-source models, we carefully select recent and commonly used models of different structures and of difference scales, such as model families of MiniCPM-V Yao et al. (2024), LLaVA (Liu et al., 2023b;c), mPLUG-Owl (Ye et al., 2024b;a), Idefics (Laurençon et al., 2024; Laurençon et al., 2024), and etc. Since the awareness of fine-grained details and instruction-aware visual features are significant indicators during evaluation, we also include models specifically optimised on these aspects, such as SEAL (Wu & Xie, 2023) for fine-grained details understanding, and Brote (Wang et al., 2024b) which is trained from InstructBLIP (Dai et al., 2023) for instruction-aware and multi-image comprehension. Please refer to Appendix D for details of all 27 evaluated models. These models are divided into two types, **single-image models** that accepting only one image per input, such as LLaVA-1.6 (Liu et al., 2023b) and MiniCPM-Llama3-V-2.5 (Yao et al., 2024); and **multi-image models** that allow more than one images to appear in the same input, such as Brote and Idefics. We will describe the approaches for integrating multiple views into the input for the two types of models in § 4.3 and Appendix G.2.

## 4.2 EVALUATION PIPELINES

We design three pipelines for different abilities, two fundamental independent abilities, zooming and shifting, and a mixture of them. Generally, the images are split into 4 sub-views to enable zooming and shifting operations, and to ensure that all models are fairly evaluated without bias from their training data or image processing strategies. We also discuss different splittings in Appendix G.1.

**Zooming pipeline.** It evaluates the ability to locate and determine fine-grained information that are necessary to answer the question. As illustrated in Figure 4 (a), this pipeline contains two stages, the view selection and the question answering stages. To simulate the zooming operation, models are required to first select sub-views to be zoomed given the initial view, then answer questions based on these zoomed views. The initial view used in this pipeline is the full image with size $w \times h$. Each of the selected sub-views will be resized to size $w \times h$, the same size as the initial view. As demonstrated in Figure 4 (c), the zoomed right-upper view is resized as a $w \times h$ image, and so does the zoomed left-lower view. Afterwards, models answer the question given the two zoomed views. The detailed prompt template are provided in Appendix J.2.

**Shifting pipeline.** It evaluates the ability to shift perceptual fields for missing information and to deduce the answer given perceived perceptual fields following templates in Appendix J.3. This is also a two-stage pipeline as shown in Figure 4 (b). To simulate the movement of human eyes, models are presented with an initial view, of size $w \times h$, which is a cropped field from the original image, and are asked to determine if the current view or views are sufficient for answering the questions. Upon receiving positive responses, models are prompted to produce answer given the current view or views. If the model requires more views to infer the answer, an adjacent view will be given until the model is able to answer the question. For this pipeline, we assign different difficulties according to human-annotated visual clues, described in § 3.2, contained in the initial views as follows:

- Shifting-R: starting the missing view examination with randomly selected initial views.
- Shifting-E: easy-level evaluation, where initial views contain at least one entire visual clue for answering the question.
- Shifting-M: medium-level evaluation, where initial views contain only partial visual clues for answering the question.
- Shifting-H: hard-level evaluation, where no visual clues appear in the initial views.

**Mixed pipeline.** While the above pipelines allow for either zooming or shifting individually, we also implement a mixed setting that does not specify the type of active perception ability required. As illustrated in Figure 4 (c), models must independently decide whether to zoom and/or shift to different perceptual fields. Unlike the zooming pipeline, where the model answers questions based on all selected views, in the mixed pipeline, a view would be discarded after zooming if the model recognizes it as irrelevant to the question. In comparison to the shifting pipeline, the mixed pipeline also provides access to the full image view in addition to cropped sub-views. Please refer to Appendix J.4 for employed prompt templates. This type of evaluation requires models to account for all the sub-views and the full image for unbiased operation determination and view selection. Otherwise, it is at risk of reverting to either zooming or shifting evaluation without sufficient and unconverted visual information. Therefore, the mixed pipeline is only applied to multi-image models during evaluation.

In addition to these pipelines for evaluating active perception abilities, our benchmark also supports the general VQA evaluation, where models are prompted to answer visual questions given original images without zooming or shifting. The detailed prompt templates are provided in Appendix J.1. We also evaluate the performance of this setting to assess the difficulty of our created benchmark.

## 4.3 PROCESSING OF VIEWS

The question answering stage of the zooming, shifting, and mixed pipelines, as well as the missing view examination stage of the shifting pipeline, require multi-image inputs if multiple views are selected. In this paper, we primarily focus on the interleaved multi-image setting, since it is more practical and natural compared to the single-image setting. Multi-image models can naturally read and understand multiple views at one time (in the form of different images) during evaluating, and we directly format the images and text in an interleaved format. However, we also propose methods for evaluating powerful single-image models. For these models, we employ two strategies to enable simultaneous understanding of different views. One is to concatenate the required views into a single

Table 2: Results of individual abilities required by active perception, following shifting and zooming pipelines in §4.2. We list results of 20 widely-discussed models here, and refer readers to Table 8 for more detailed results. The humane performance is **84.67%** referring to Table 5 in Appendix B. The two gray columns, "Full image" and "Single View", are provided only as references for general QA without active perception. "Model AVG": average scores of zooming (column "Zooming") and shifting (columns "Shifting-R", "Shifting-E", "Shifting-M", "Shifting-H") evaluations. The best scores of each column are **bolded** and the best scores in each model types are highlighted .

| Models | Zooming Ability | | Shifting Ability | | | | | Models AVG |
|---|---|---|---|---|---|---|---|---|
| | Full image | Zooming | Single View | Shifting-R | Shifting-E | Shifting-M | Shifting-H | |
| *Proprietary models (APIs)* | | | | | | | | |
| Gemini-1.5-pro | **73.85** | **72.31** | 58.15 | **67.08** | **67.38** | 65.54 | **67.69** | **68.00** |
| GPT-4o | 67.38 | 68.62 | **61.23** | **67.08** | 66.77 | 65.23 | 64.31 | 65.54 |
| Claude 3.5 Sonnet | 72.92 | 71.69 | 54.46 | 65.23 | 66.15 | 60.31 | 61.85 | 65.05 |
| *Open-source models for multiple images as input* | | | | | | | | |
| Qwen2-VL | 63.08 | 64.62 | 54.46 | 61.23 | 62.77 | 64.31 | 61.85 | 62.96 |
| Idefics3-8B-Llama3 | 59.08 | 58.15 | 53.23 | 61.85 | 59.38 | 59.69 | 60.31 | 59.88 |
| MiniCPM-V 2.6 | 64.62 | 61.85 | 54.46 | 54.77 | 61.23 | 58.15 | 55.69 | 58.34 |
| mPLUG-Owl3 | 62.46 | 60.92 | 54.15 | 51.69 | 56.31 | 55.69 | 53.54 | 55.63 |
| LLaVA-OneVision | 64.92 | 65.23 | 56.92 | 53.54 | 57.23 | 52.31 | 48.62 | 55.39 |
| InternVL2-8B | 58.15 | 56.00 | 45.85 | 54.77 | 59.70 | 53.23 | 52.00 | 55.14 |
| idefics2-8b | 61.85 | 61.85 | 55.69 | 53.23 | 56.92 | 51.69 | 49.23 | 54.58 |
| Mantis | 59.08 | 60.62 | 52.92 | 52.92 | 55.38 | 52.92 | 52.31 | 54.83 |
| Brote-IM-XL-3B | 54.77 | 54.46 | 55.69 | 51.38 | 51.08 | 52.62 | 47.69 | 51.45 |
| MMICL-XXL-11B | 51.69 | 49.54 | 50.15 | 49.85 | 49.85 | 46.77 | 45.54 | 48.31 |
| *Open-source models for single image as input* | | | | | | | | |
| MiniCPM-Llama3-V-2.5 | 63.87 | 61.25 | 54.47 | 60.92 | 60.31 | 59.38 | 58.46 | 60.06 |
| GLM-4V-9B | 67.08 | 56.92 | 53.85 | 56.92 | 60.62 | 56.00 | 52.92 | 56.68 |
| InternVL-Vicuna-13B | 56.92 | 62.77 | 52.31 | 53.85 | 52.92 | 52.92 | 51.08 | 54.71 |
| LLaVA-1.6 7B | 55.08 | 68.92 | 50.15 | 51.69 | 52.31 | 49.23 | 48.00 | 54.03 |
| mPLUG-Owl2-7B | 55.08 | 55.38 | 52.00 | 47.38 | 46.46 | 46.46 | 46.15 | 48.37 |
| Mini-Gemini-7B-HD | 55.69 | 34.77 | 51.70 | 48.62 | 48.00 | 47.69 | 50.15 | 45.85 |
| SEAL | 48.31 | 54.77 | 42.77 | 42.15 | 42.77 | 40.02 | 40.62 | 44.07 |

Table 3: Results of mixed pipeline for multi-image models. "ACC": accuracy. "#zoom": average zooming operations. "#shift": average shifting operations. "#view": average used views.

| GPT-4o | | | | Qwen2-VL | | | | MiniCPM-V 2.6 | | | | mPLUG-Owl3 | | | | Idefics3 | | | |
|---|---|---|---|---|---|---|---|---|---|---|---|---|---|---|---|---|---|---|---|
| ACC | #zoom | #shift | #view | ACC | #zoom | #shift | #view | ACC | #zoom | #shift | #view | ACC | #zoom | #shift | #view | ACC | #zoom | #shift | #view |
| 69.54 | 1.61 | 1.23 | 1.35 | 65.54 | 2.51 | 2.17 | 2.12 | 64.00 | 1.31 | 0.39 | 0.94 | 59.69 | 2.59 | 1.49 | 1.43 | 62.15 | 1.16 | 0.59 | 0.58 |

flattened image, and the other preserves merely the current view as an image while converting the remainings into textual descriptions. Please refer to Appendix G.2 for details.

## 5 RESULTS AND ANALYSIS

**Main results.** We adopt accuracy as the evaluation metric for question answering. Experimental results of zooming and shifting pipelines on our benchmark are listed in Table 2, whose elaborated results on each type are provided in Appendix F. Results of human evaluations are discussed in detail in Appendix B. We obtain the random choice result of 33.95%, averaged over 10k runs.

Firstly, we can conclude from Table 2 that all the evaluated models perform better than random guessing, indicating their potential to maintain active perception abilities of zooming and shifting. However, even the average results of powerful proprietary models are still much lower then human performance (84.67%, as in Table 5). Secondly, although proprietary models achieve better overall performances compared to open-source models, the performance gap between these two categories are considerably smaller compared to gaps observed in other tasks from previous research. With our evaluation pipelines, proprietary models achieve the highest average score of 68.00%, whereas the highest from open-source models is 62.96%. Thirdly, among open-source models, multi-image models present better average results compared to single-image models, particularly in shifting evaluations where models can only access constrained views.

For results of mixed evaluation in Table 3, we notice that the evaluated models benefit from enabling complex active perception and outperform individual zooming or shifting settings on average. Among the evaluated models, MiniCPM-V 2.6 (64.00%) and Idefics3-8B-Llama3 (62.15%) achieve even higher accuracy compared to the results of providing human-annotated ground truth views from Table 4 (62.77% and 60.92%, respectively). The mixed pipeline encourages models to zoom and/or shift perceptual fields autonomously, similar to human behavior, and these results demonstrate the significance and effectiveness of active perception abilities. However, during experiments, we observe that some multi-image models fail to follow instructions of mixed evaluation by generating irrelevant responses or selecting invalid views, which, unfortunately, disrupts the mixed process.

**Impacts of selected views.** Our evaluation pipelines involve selecting useful view in their first stages. The reliability of selected views plays a crucial role in the following question answering stage. We compute the recall of used views following Equation 1 in Appendix E, and include results in Table 4, along with accuracy of providing models with groundtruth views. Overall, higher selection recall tends to correlate with higher question answering accuracy. For example, Idefics3-8B-Llama3 and InternVL2-8B present the lowest recalls (41.09%) among multi-image models in Table 4, leading to lower zooming evaluation accuracies of 56.00% and 58.15%, respectively. For the shifting evaluation, some models keep shifting view until all four views are inquired. However, this does not necessarily support a better accuracy, as some views contain redundant informa-

Table 4: Results of view selection ("$R_{select}$") and accuracy when given groundtruth views ("$ACC_{GT}$", 2.64 views on average) that contain human-annotated visual clues. "$ACC_{QA}$": accuracy of question answering for zooming and shifting. "#view": average counts of selected views.

| Models | $ACC_{GT}$ | Zooming | | | Shifting-R | | |
|---|---|---|---|---|---|---|---|
| | | $ACC_{QA}$ | $R_{select}$ | #view | $ACC_{QA}$ | $R_{select}$ | #view |
| *Multi-image Models* | | | | | | | |
| GPT-4o | 73.54 | 68.62 | 69.03 | 2.29 | **67.08** | 60.54 | 3.26 |
| mPLUG-Owl3 | 60.62 | 60.92 | 68.57 | 2.66 | 51.69 | **74.62** | 4.00 |
| Claude 3.5 Sonnet | 72.31 | 71.69 | 67.64 | 2.19 | 65.23 | 45.52 | 2.47 |
| Qwen2-VL | 65.85 | 64.62 | 64.61 | 2.35 | 61.23 | **74.62** | 4.00 |
| Gemini-1.5-pro | 72.00 | **72.31** | 62.63 | 2.10 | **67.08** | 46.33 | 2.46 |
| MiniCPM-V 2.6 | 62.77 | 61.85 | 57.03 | 2.20 | 54.77 | 54.83 | 2.98 |
| LLaVA-OneVision | 64.92 | 65.23 | 46.67 | 2.35 | 53.54 | 37.14 | 2.02 |
| Idefics3-8B-Llama3 | 60.92 | 58.15 | 41.09 | 1.52 | 61.85 | **74.62** | 4.00 |
| InternVL2-8B | 73.23 | 56.00 | 41.09 | 1.53 | 54.77 | 45.75 | 2.61 |
| *Single-image Models* | | | | | | | |
| InternVL-Vicuna-13B | 68.00 | 62.77 | **83.47** | 3.31 | 53.85 | 69.73 | 3.75 |
| LLaVA-1.6 13B | 67.69 | 68.92 | 68.57 | 2.65 | 51.69 | **74.62** | 4.00 |
| SEAL | 56.92 | 54.77 | 68.22 | 2.74 | 42.15 | 71.48 | 3.83 |
| MiniCPM-Llama3-V-2.5 | 62.20 | 61.25 | 66.12 | 2.46 | 53.85 | 63.56 | 3.42 |
| mPLUG-Owl2-7B | 67.38 | 55.38 | 47.61 | 1.97 | 47.38 | **74.62** | 4.00 |
| GLM-4V-9B | **74.46** | 56.92 | 30.62 | 1.08 | 56.92 | 50.17 | 2.64 |

tion that might distract the model during reasoning. Additionally, we observe shifting evaluations tend to require more views to be used for answering questions than zooming evaluations, yet it often results in inferior overall performance compared to zooming. This is because some of the current advanced models struggle to either move their field of views for necessary visual details, or screen out distracting information. Therefore, we believe that more attention should be paid to evaluating and enhancing active perception abilities of MLLMs given constraint perceptual fields.

**Analysis of difficulties for shifting evaluation.** Generally, the accuracy of question answering and the recall of view selection decrease as the difficulties of the initial views increases. As shown by typical results for LLaVA-OneVision and GLM-4V-9B in Table 2, the gaps between shifting-E and shifting-H are as large as 8.61% and 7.70%, respectively. However, we observe some exceptions from Gemini-1.5-pro and Idefics3, where the accuracy of shifting-H is slightly higher than shifting-E. For Gemini-1.5-pro, the recall for shifting-H is 47.73%, over 1 point higher than shifting-E (45.29%). We conclude that Gemini-1.5-pro achieves higher accuracy on shifting-H due to the acquisition of more proper views. However, Idefics3 presents a different trend. It maintains a recall of 74.64% from shifting-E to shifting-H, but achieves a higher accuracy on shifting-H. We hypothesis that the performance gain of this model comes from the order of input views, where hard-level evaluation starts with less relevant views and appends more useful views at the end of the image sequence when all the views are selected.

**Analysis of view splitting and image processing strategies.** We investigate these two aspects in Appendix G.1 and Appendix G.2, respectively. For the splitting settings, the adopted 4 sub-image setting provides fair and reliable evaluation results, which is not only effective and efficient, but also demonstrate a good balance between zooming and shifting evaluations. For the strategy of converting image into text, on the contrary, we observe significant drops of results on both zooming and shifting evaluations for most of investigated models. This suggests that the resizing issue in image concatenation strategy has only a minor impact on the performance. Please refer to Appendix G.1 and Appendix G.2 for details.

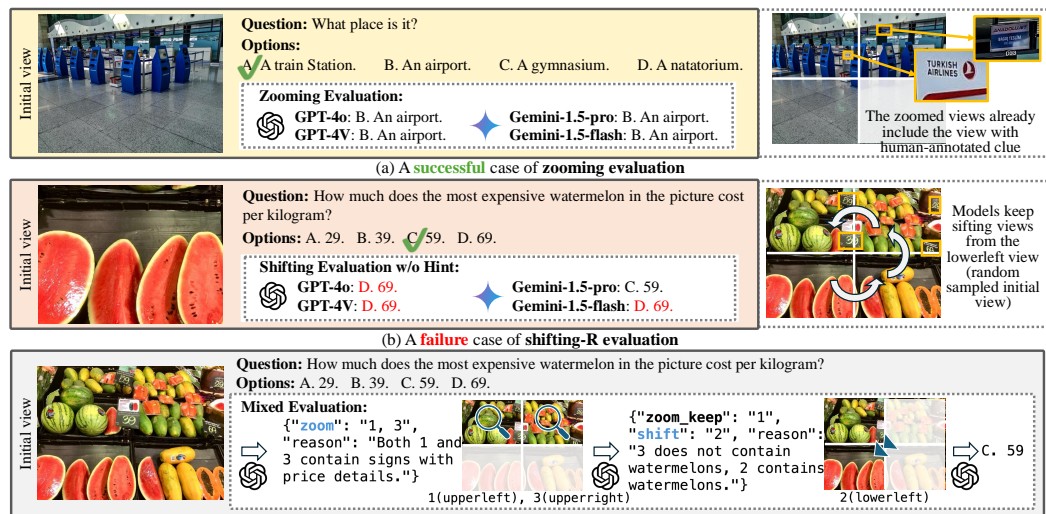

Figure 5: Cases for each evaluation pipelines. (a) a succeeded **zooming** case, (b) a failed **shifting** case, and (c) a **mixed** case that successfully corrects the wrong answer produced by (b). Model selected views for case (a) and (b) are placed to the right of example frames, and used views for case (c) are shown with in its frame as the selection of views changes during the evaluation process.

**Analysis of Cases**   We demonstrate three cases for the three proposed pipeline in Figure 5, and provide additional examples of integrating human-annotated visual clues hints in Appendix H. These results are generated by GPT-4 models and Gemini-1.5 models. Case (a) stands for the zooming evaluation, where models successfully identify the view containing useful information and generate the correct result. Case (b) illustrates a failure in the shifting-R evaluation, where all the models continue shifting to new views until all views are used. Though including the correct views, the additional views severely distract the reasoning process, where three out of four employed models produce incorrect answers. To explore how human-like mixed evaluation affects the visual reasoning process, we further exam this failure case using GPT-4o. As shown in Figure 5 case (c), GPT-4o first zooms into the "upper left" and "upper right" views, then discards the "upper right" view and shifts to the "lower left" one, which finally leads to the correct answer. Notably, in the final preserved views, distracting information (the highest price tag on papaya, "69") is screened out. This indicates that GPT-4o exhibits decent active perception abilities to move the field of view, locate details, and filter out distracting information

## 6   CONCLUSION AND FUTURE WORK

This paper introduces ActiView, a novel benchmark designed to evaluate the active perception abilities of MLLMs. ActiView simulates real-world scenarios by imposing view constraints on images, requiring models to perform view shifting and/or zooming to gather necessary information for answering questions. Our results indicate that current MLLMs exhibit significantly lower active perception capabilities compared to humans, and that active perception abilities of models will be markedly enhanced by allowing inputs in multi-image interleaved structures. We also observed that models tend to perform better on our zooming evaluations compared to shifting evaluations. This suggests that the evaluated models lack the ability to combine their understandings of constrained perceptual fields to form a holistic perspective of the complete image or the full scene. We hope our benchmark will inspire further research in this critical area.

In this study, we assess the active perception abilities of models through question answering. While this approach presents significant challenges for current MLLMs, it does not encompass all aspects of active perception. For instance, techniques like tool learning and multi-agent collaboration could also be investigated concerning the factors such as perspective distortion, multi-sensor integration, and the incorporation of more dynamic or interactive environments. Given that these limitations exceed the scope of a single paper, we will leave them as future work.

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

## A   DATA COLLECTION AND ANNOTATION

### A.1   IMAGE COLLECTION

To ensure the clearness of useful visual details, the collected original image should be of high resolution. In practice, we collected images of three resolution levels, including $1920 \times 1040$, $2250 \times 1500$, and $5184 \times 3456$, which are originated from VCR dataset (Zellers et al., 2019), SA-1B dataset (Kirillov et al., 2023), and photos taken in daily life. At the beginning of the image collection process, 30 images are collected from photos taken from daily activities, which are then served as pilots and standards for manually expanding the data scale from SA-1B dataset (Kirillov et al., 2023) and VCR dataset (Zellers et al., 2019). These images should include rich and fine-grained visual details.

### A.2   RULES FOR ANNOTATION

We provide a concise version of instruction used during annotation. For each of the images, annotators should follow the following instructions:

- Questions: (1) Questions should be objective which have one and only one answer regarding the images. (2) The participation of multiple visual clues are preferred. They can be in the same or different regions of the image.
- Options: (1) Options originated from the image itself are preferred. (2) The numeric options should be arrange randomly, neither descending nor ascending order. (3) Options cannot be opposite to each other, except for "Yes" or "No". (4) The number of options are not restricted to 4, you can provide as many options as long as they are reasonable and are closely related to the question and the image.
- Distraction: Annotator should provide distracting visual clues that could lead to wrong answer (if any).

## B   HUMAN EVALUATION

We sampled 60 questions for human-level test, and recruit 6 testees, who did not participate in image collection and question annotation, to evaluate the human level performance of our benchmark. These testees are from diverse backgrounds, including computer science (CS), telecommunication (Telecom), Medicine (Med), and Art.

For a fair comparison with the MLLMs, we employ two settings, including a "Human" evaluation that asks testees to answer questions all by themselves, and a "Human*" evaluation that allows testees to use the Internet and LLMs for the required knowledge, because these testees may not be exposed to knowledge that never appear in their everyday life, which MLLMs have already seen in the training data. Note that in the "Human*" evaluation, directly search for

Table 5: Human level performance and question consistency. Consis.: human-annotated consistency of question-image-option-groundtruth. ACC: accuracy of answering the questions without assistance (*i.e.*, the accuracy for "Human" evaluation). ACC*: accuracy of answering the questions with the help of Internet (*i.e.*, the accuracy for "Human*" evaluation).

| Annotator | Background | ACC | ACC* | Consis. |
|---|---|---|---|---|
| User1 | CS | 73.33 | 85.00 | 100.00 |
| User2 | Med | 71.67 | 75.00 | 93.33 |
| User3 | Telecom | 85.00 | 90.00 | 100.00 |
| User4 | CS | 81.67 | 88.33 | 83.33 |
| User5 | CS | 76.67 | 85.00 | 95.00 |
| User6 | Art | 70.00 | 78.83 | 83.33 |
| Average | | 77.53 | 84.67 | 94.20 |

the answer to the questions are forbidden. Referring to the question in Figure 4 as an example, testees may search for "*what does the national flag of UK/France/Spain/Hungary look like?*", which may provide extra knowledge that helps them to answer the original question. Manual evaluation achieves an average accuracy of 84.67%, which is more than doubled of the random choice result (33.95%), while some models present only slightly higher accuracies compared to the random result. These indicate the potential for models to get improved.

We ask testees to vote for the consistency of the annotated question-image-option-groundtruth quadruple for the investigation of the reliability of our benchmark. The consistency score represents

if the testee agree with these quadruples and find the groundtruth answers and the provided options are practical and reasonable. Our benchmark is reliable indicated by a consistency of 94.2%.

## C  TEXT-ONLY EVALUATION

We provide a text-only evaluation to measure the amount of commonsense answers with providing images in our benchmark. We conducted two experiments:

- Commonsense-only evaluation. This evaluation aims at measuring the amount of questions that can be answered only via commonsense knowledge without searching for visual clues in the image. The template is as follows: "Please answer questions based on you commonsense knowledge. If you are not able to answer the question based soly on the commonsense knowledge you've acquired, please **response with 'None'**. Question Options Your answer:"
- Commonsense and data bias evaluation. Considering that current models are trained with a large amount of data and various tasks, they could potentially memories the most frequent answers given a image-question pair. We implement another template to evaluate the amount of data that can be correctly guessed without corresponding context. The template is as follows: "Please answer questions based on you commonsense knowledge. If you are not able to answer, please **select a most probable one**. Question Options Your answer:"

Table 6: Results of text-only evaluation. ACC: answer with commonsense only without random guessing. ACC(guess): guess the answer according to commonsense.

| Model | Claude | GPT-4o | Qwen2-VL | MiniCPM-V 2.6 | Idefics3 | Brote-IM-XL |
|---|---|---|---|---|---|---|
| ACC | 2.14 | 2.45 | 23.38 | 26.77 | 44.92 | 40.00 |
| ACC(guess) | 26.07 | 37.73 | 42.77 | 41.54 | 47.38 | 40.00 |

Results for these text-only evaluations are listed in Table 6. This table indicates that questions in our benchmark cannot be simply answered via commonsense, where two powerful models GPT-4o and Claude achieves only 2.45% and 2.14% for commonsense-only evaluation. The row of ACC(guess) presents results of generating the most probable answers, reflecting the bias obtained from the training corpus. The differences between these two type of evaluation are caused by the ability of instruction-following. We found that Idefics3 and Brote-IM-XL present weaker instruction-following ability compared to other models in this table, that they still exhibit a behavior of guessing when commonsense cannot be used to answer the questions.

Overall, our benchmark requires elaborate observation of the given images and comprehensive understanding of image-question pairs, which cannot be solved simply by commonsense.

## D  MODELS

Our evaluated APIs include GPT-4V (OpenAI, 2023), GPT-4o (OpenAI, 2024), and Gemini-1.5-pro (Reid et al., 2024). For open-source models, we include GLM-4V (Du et al., 2022), SEAL (Wu & Xie, 2023), Mini-Gemini (Li et al., 2024c), MMICL (Zhao et al., 2024), InternVL series (Chen et al., 2023; 2024), Brote (Wang et al., 2024b), LLaVA-1.6 (Liu et al., 2024), LLaVA-OneVision (Li et al., 2024a), mPLUG-Owl series (Ye et al., 2024b;a), MiniCPM-Llama3-V-2.5, MiniCPM-V 2.6, Qwen2-VL (Wang et al., 2024a). Details of these models are listed in Table 7. Considering models of different scales, we include a total of 27 models.

## E  CALCULATION OF VIEW SELECTION RECALL

We follow the recall metric to evaluate the performance of the view selection for zooming setting and the missing view examination for shifting settings. We denote the selected views containing human-annotated clues as $TP_{op}$, where $op$ refers to either "zoom", "shift" or "mix". $FN_{op}$ refers to views that contain human-annotated clues but are not selected for answering questions. Finally, the

Table 7: The versions of LLM backbone and vision encoder of our evaluated models. For proprietary models, we provide the API version we used.

| Models | LLM Backbone | Vision Encoder |
|---|---|---|
| *APIs* | | |
| GPT-4o (OpenAI, 2024) | gpt-4o | |
| Gemini-1.5-pro (Reid et al., 2024) | gemini-1.5-pro | |
| Claude 3.5 Sonnet (Anthropic, 2024) | claude-3-5-sonnet-20240620 | |
| *Open-Source Models* | | |
| GLM-4V-9B (Du et al., 2022) | GLM-4-9B | CLIP |
| MiniCPM-Llama3-V-2.5 (Yao et al., 2024) | Llama-3-8B | SigLip-400M |
| MiniCPM-V 2.6 (Yao et al., 2024) | Qwen2-7B | SigLip-400M |
| SEAL (Wu & Xie, 2023) | Vicuna-7B | CLIP ViT-L/14 |
| LLaVA-1.6-13B (Liu et al., 2024) | Vicuna-13B | CLIP-ViT-L/14 |
| LLaVA-1.6-7B (Liu et al., 2024) | Vicuna-7B | CLIP-ViT-L/14 |
| LLaVA-OneVision-7B (Li et al., 2024a) | Qwen2-7B | SO400M |
| mPLUG-Owl2-7B (Ye et al., 2024b) | Llama-2-7B | CLIP ViT-L/14 |
| mPLUG-Owl3-7B (Ye et al., 2024a) | Qwen2-7B | Siglip-400m |
| InternVL-Vicuna-7B (Chen et al., 2023) | Vicuna-7B | InternViT |
| InternVL-Vicuna-13B (Chen et al., 2023) | Vicuna-13B | InternViT |
| InternVL-Vicuna-13B-448px (Chen et al., 2023) | Vicuna-13B | InternViT-300M-448px |
| InternVL2-8B (Chen et al., 2024) | internlm2_5-7b-chat | InternViT-300M-448px |
| Qwen2-VL-8B (Wang et al., 2024a) | Qwen2-7B | OpenCLIP-ViT-bigG |
| Mantis (Jiang et al., 2024) | LLaMA-3 | Siglip-400m |
| Idefics2-8B (Laurençon et al., 2024) | Mistral-7B | Siglip-400m |
| Idefics2-8B-base (Laurençon et al., 2024) | Mistral-7B | Siglip-400m |
| Idefics3-8B-Llama3(Laurençon et al., 2024) | Mistral-7B | Siglip-400m |
| MMICL-XXL (Zhao et al., 2024) | FlanT5-XXL-11B | EVA-G |
| Brote-IM-XXL (Wang et al., 2024b) | FlanT5-XXL-11B | EVA-G |
| MMICL-XL (Zhao et al., 2024) | FlanT5-XL-3B | EVA-G |
| Brote-IM-XL (Wang et al., 2024b) | FlanT5-XL-3B | EVA-G |
| Mini-Gemini-7B-HD (Li et al., 2024c) | LLaMA-3 | CLIP-L |
| Mini-Gemini-7B (Li et al., 2024c) | LLaMA-3 | CLIP-L |

$R_{\text{select}}$ is calculated as follows:

$$R_{\text{select}} = \frac{TP_{op}}{TP_{op} + FN_{op}}, \quad op \in \{\text{zoom}, \text{shift}, \text{mix}\}. \tag{1}$$

The view selection recalls of selected models are listed in Table 4.

# F    EXPERIMENTAL RESULTS

We reported the full results of 27 models in Table 8. This table preserves the conclusions as discussed in by Table 2. The detailed results of each categories are listed in Table 9, Table 10, Table 11, Table 12, and Table 13, for zooming, shifting-R, shifting-E, shifting-M, and shifting-H, respectively.

## F.1    ANALYSIS OF RESULTS ON ZOOMING EVALUATION

We notice that for the zooming evaluation, except for InternVL and LLaVA-1.6, single-image models fail to achieve equivalent or comparable results (comparing with full image setting), and present performance gap of as large as 29.23% (for Mini-Gemini-7B) where the zooming results are much lower. These imply that some single models are unaware of the location of key visual information required by the target question. On the contrary, multi-image models present comparable or even better scores under the zooming evaluation.

We summarise the zooming results on sub-classes from Table 9, that the environment-centric category (including Geo-Loc, Orient, and Daily-Loc) presents significantly higher scores than object-centric and event-centric categories. The reason lies in the fact that questions in environment-centric category require more visual commonsense that most of models learnt from the vast training data. We also notice that Idefics2-8B-base even enlarges the performance gap between environment-centric category and the others by around 40%, which demonstrate extremely unbalanced capabilities of exploiting inherent commonsense and observed visual clues. The most challenging types of instances are Orient, Count-Dis and Event-S, that present even halved scores compared to the other sub-classes. Surprisingly, some of evaluated single-image models achieve better scores or perform

Table 8: The evaluation of active perception abilities on our benchmark, including zooming (for limited resolution scenarios), and shifting (for scenarios of limiting the field of views). "Model AVG": average scores of column "Zooming", "Shifting-R", "Shifting-E", "Shifting-M", and "Shifting-H". The best scores of each column are **bolded** and the best scores in each model types are highlighted .

| Models | Zooming | | Shifting | | | | | Models AVG |
|---|---|---|---|---|---|---|---|---|
| | Full image | Zooming | Single View | Shifting-R | Shifting-E | Shifting-M | Shifting-H | |
| *proprietary models* | | | | | | | | |
| Gemini-1.5-pro | **73.85** | **72.31** | 58.15 | **67.08** | **67.38** | 65.54 | **67.69** | **68.00** |
| GPT-4o | 67.38 | 68.62 | **61.23** | **67.08** | 66.77 | 65.23 | 64.31 | 66.40 |
| Claude 3.5 Sonnet | 72.92 | 71.69 | 54.46 | 65.23 | 66.15 | 60.31 | 61.85 | 65.05 |
| *Open-source models for multiple images as input* | | | | | | | | |
| Qwen2-VL | 63.08 | 64.62 | 54.46 | 61.23 | 62.77 | 64.31 | 61.85 | 62.96 |
| Idefics3-8B-Llama3 | 59.08 | 58.15 | 53.23 | 61.85 | 59.38 | 59.69 | 60.31 | 59.88 |
| MiniCPM-V 2.6 | 64.62 | 61.85 | 54.46 | 54.77 | 61.23 | 58.15 | 55.69 | 58.34 |
| mPLUG-Owl3 | 62.46 | 60.92 | 54.15 | 51.69 | 56.31 | 55.69 | 53.54 | 55.63 |
| LLaVA-OneVision | 64.92 | 65.23 | 56.92 | 53.54 | 57.23 | 52.31 | 48.62 | 55.39 |
| InternVL2-8B | 58.15 | 56.00 | 45.85 | 54.77 | 59.70 | 53.23 | 52.00 | 55.14 |
| Mantis | 59.08 | 60.62 | 52.92 | 52.92 | 55.38 | 52.92 | 52.31 | 54.83 |
| Idefics2-8B | 61.85 | 61.85 | 55.69 | 53.23 | 56.92 | 51.69 | 49.23 | 54.58 |
| Brote-IM-XL-3B | 54.77 | 54.46 | 55.69 | 51.38 | 51.08 | 52.62 | 47.69 | 51.45 |
| Idefics2-8B-base | 52.62 | 48.62 | 47.69 | 49.54 | 50.77 | 47.69 | 47.69 | 48.86 |
| Brote-IM-XXL-11B | 53.85 | 54.77 | 49.23 | 49.85 | 50.77 | 44.92 | 43.69 | 48.80 |
| MMICL-XXL-11B | 51.69 | 49.54 | 50.15 | 49.85 | 49.85 | 46.77 | 45.54 | 48.31 |
| MMICL-XL-3B | 49.85 | 49.85 | 44.31 | 44.92 | 48.92 | 45.85 | 44.31 | 46.77 |
| *Open-source models for single image as input* | | | | | | | | |
| MiniCPM-Llama3-V-2.5 | 63.87 | 61.25 | 54.47 | 60.92 | 60.31 | 59.38 | 58.46 | 60.06 |
| GLM-4V-9B | 67.08 | 56.92 | 53.85 | 56.92 | 60.62 | 56.00 | 52.92 | 56.68 |
| InternVL-Vicuna-13B | 56.92 | 62.77 | 52.31 | 53.85 | 52.92 | 52.92 | 51.08 | 54.71 |
| LLaVA-1.6 7B | 55.08 | 68.92 | 50.15 | 51.69 | 52.31 | 49.23 | 48.00 | 54.03 |
| InternVL-Vicuna-7B | 55.38 | 65.23 | 51.70 | 52.92 | 51.38 | 50.77 | 48.62 | 53.78 |
| LLaVA-1.6 13B | 56.92 | 65.23 | 52.31 | 45.85 | 55.08 | 52.62 | 48.92 | 53.54 |
| InternVL-Vicuna-13B-448px | 50.46 | 57.85 | 45.54 | 48.31 | 48.31 | 48.92 | 48.92 | 50.46 |
| mPLUG-Owl2-7B | 55.08 | 55.38 | 52.00 | 47.38 | 46.46 | 46.46 | 46.15 | 48.37 |
| Mini-Gemini-7B-HD | 55.69 | 34.77 | 51.70 | 48.62 | 48.00 | 47.69 | 50.15 | 45.85 |
| SEAL | 48.31 | 54.77 | 42.77 | 42.15 | 42.77 | 40.02 | 40.62 | 44.07 |
| Mini-Gemini-7B | 47.08 | 17.85 | 47.38 | 39.38 | 38.15 | 38.15 | 36.00 | 33.91 |

equally compared to powerful proprietary models for the zooming evaluation, especially mPLUG-Owl2-7B regarding the object-centric category. We hypothesis that this model possesses strong object recognition ability and is less affected by object hallucination compared to other MLLMs.

## F.2  ANALYSIS OF RESULTS ON SHIFTING EVALUATION

Results of shifting-R evaluation are shown in Table 10, and the level-specified shifting evaluation are listed in Table 11, Table 12 and Table 13. Similar to that of zooming evaluation, results on environment-centric category are significantly better than the ones on object-centric and event-centric categories. The results of proprietary models are better than the results of open-source models, and that models for multiple images perform better than models for single image. We observe a trend where, as the difficulty increases, the superiority of open-source multi-image models becomes more evident.

There is overall trend for all the sub-classed that the accuracy decreases as the difficulty is getting increased. However, exceptional performances are identified for Gemini-1.5-pro, Idefics3-8B-Llama3, and Mini-Gemini-7B-HD, where the results of shifting-H even outperform the results of shifting-E. There is a potential reason that the hard-level evaluation starts with less relevant views and appends more useful views at the end of the image sequence. The performance of these models are more significantly influenced by the order of presented images compared to the rest models. The degradation of performance is more remarkable for the environment-centric and the object-centric categories compared to the event-centric category. Regarding the increasing of difficulty for the environment-centric and the object-centric categories, we observe gaps of about 10% for models such as LLaVA-OneVision, Idefics2-8B, Brote, MMICL, GLM-4V-9B and Mini-Gemini-7B. These observations indicate that different initial perceptual fields have distinct impacts on instances that requiring demanding attention on subtle changes of fine-grained objects. Results show that GPT-4o consistently outperforms other models in the average score of the environment-centric category, implying robust event capture and understanding capabilities in multi-image scenarios.

Table 9: Results on sub-classes of zooming evaluation.

| Models | Type I | | | AVG | Type II | | | AVG | Type III | | AVG |
|---|---|---|---|---|---|---|---|---|---|---|---|
| | Geo-Loc | Orient | Daily-Loc | | Obj-Attr | Obj-Rel | Count-Dis | | Event-M | Event-S | |
| *proprietary models* | | | | | | | | | | | |
| Gemini-1.5-pro | 91.89 | 60.00 | **92.68** | **83.33** | 80.00 | 65.22 | **51.06** | 65.73 | **85.29** | 57.50 | 70.27 |
| GPT-4o | 94.59 | 63.33 | 85.37 | 82.41 | 68.00 | 54.35 | 46.81 | 56.64 | 76.47 | 65.00 | 70.27 |
| Claude 3.5 Sonnet | **97.30** | 50.00 | 87.80 | 80.56 | 72.00 | 67.39 | 42.55 | 60.84 | 82.35 | 75.00 | **78.38** |
| *Open-source models for multiple images as input* | | | | | | | | | | | |
| Qwen2-VL | **97.30** | 50.00 | 80.49 | 77.78 | 68.00 | 65.22 | 40.43 | 58.04 | 58.82 | 57.50 | 58.11 |
| Idefics3-8B-Llama3 | 89.19 | 56.67 | 73.17 | 74.07 | 60.00 | 54.35 | 29.79 | 48.25 | 58.82 | 50.00 | 54.05 |
| MiniCPM-V 2.6 | 86.49 | 46.67 | 80.49 | 73.15 | 54.00 | 56.52 | 31.91 | 47.55 | 61.76 | 42.50 | 51.35 |
| mPLUG-Owl3 | 89.19 | 53.33 | 80.49 | 75.93 | 64.00 | 60.87 | 36.17 | 53.85 | 58.82 | 47.50 | 52.70 |
| LLaVA-OneVision | 91.89 | 46.67 | 87.80 | 77.78 | 74.00 | 58.70 | 42.55 | 58.74 | 61.76 | 57.50 | 59.46 |
| InternVL2-8B | 75.68 | 56.67 | 70.73 | 68.52 | 60.00 | 47.83 | 25.53 | 44.76 | 61.76 | 57.50 | 59.46 |
| Mantis | 89.19 | 41.38 | 80.00 | 72.64 | 72.00 | 54.35 | 44.68 | 57.34 | 54.55 | 51.28 | 52.78 |
| Idefics2-8B | 89.19 | 63.33 | 85.37 | 80.56 | 72.00 | 50.00 | 40.43 | 54.55 | 55.88 | 45.00 | 50.00 |
| Brote-IM-XL-3B | 86.49 | 40.00 | 73.17 | 68.52 | 60.00 | 43.48 | 40.43 | 48.25 | 44.12 | 47.50 | 45.95 |
| Idefics2-8B-base | 89.19 | 56.67 | 78.05 | 75.93 | 42.00 | 39.13 | 29.79 | 37.06 | 23.53 | 35.00 | 29.73 |
| Brote-IM-XXL-11B | 86.49 | 33.33 | 80.49 | 69.44 | 58.00 | 43.48 | 34.04 | 45.45 | 58.82 | 45.00 | 51.35 |
| MMICL-XXL-11B | 67.57 | 53.33 | 65.85 | 62.96 | 52.00 | 36.96 | 34.04 | 41.26 | 58.82 | 35.00 | 45.95 |
| MMICL-XL-3B | 70.27 | 43.33 | 68.29 | 62.04 | 58.00 | 34.78 | 36.17 | 43.36 | 38.24 | 50.00 | 44.59 |
| *Open-source models for single image as input* | | | | | | | | | | | |
| MiniCPM-Llama3-V-2.5 | 86.49 | 53.33 | 75.61 | 73.15 | 64.00 | 43.48 | 31.91 | 46.85 | 50.00 | 50.00 | 50.00 |
| GLM-4V-9B | 78.38 | 53.33 | 75.61 | 70.37 | 60.00 | 45.65 | 31.91 | 46.15 | 61.76 | 55.00 | 58.11 |
| InternVL-Vicuna-13B | 72.97 | 43.33 | 85.37 | 69.44 | 68.00 | 58.70 | 29.79 | 52.45 | 73.53 | 72.50 | 72.97 |
| LLaVA-1.6 7B | 91.89 | **66.67** | 87.80 | **83.33** | 76.00 | 60.87 | 42.55 | 60.84 | 79.41 | 65.00 | 71.62 |
| InternVL-Vicuna-7B | 86.49 | **66.67** | 82.93 | 79.63 | 64.00 | 65.22 | 42.55 | 57.34 | 67.65 | 52.50 | 59.46 |
| LLaVA-1.6 13B | 94.59 | 56.67 | 90.24 | 82.41 | 78.00 | 69.57 | 36.17 | 61.54 | 64.71 | **77.50** | 71.62 |
| InternVL-Vicuna-13B-448px | 48.65 | 53.33 | 63.41 | 55.56 | 74.00 | 58.70 | 36.17 | 56.64 | 64.71 | 62.50 | 63.51 |
| mPLUG-Owl2-7B | 91.89 | 60.00 | 90.24 | 82.41 | **84.00** | **71.74** | **51.06** | **69.23** | 70.59 | 70.00 | 70.27 |
| Mini-Gemini-7B-HD | 62.16 | 26.67 | 26.83 | 38.89 | 26.00 | 43.48 | 27.66 | 32.17 | 44.12 | 25.00 | 33.78 |
| SEAL | 70.27 | 46.67 | 63.41 | 61.11 | 64.00 | 50.00 | 44.68 | 53.15 | 41.18 | 55.00 | 48.65 |
| Mini-Gemini-7B | 37.84 | 16.67 | 21.95 | 25.93 | 6.00 | 17.39 | 8.51 | 10.49 | 20.59 | 20.00 | 20.27 |

Table 10: Results on each sub-classes of Shifting-R, shifting with random initial views.

| Models | Type I | | | AVG | Type II | | | AVG | Type III | | AVG |
|---|---|---|---|---|---|---|---|---|---|---|---|
| | Geo-Loc | Orient | Daily-Loc | | Obj-Attr | Obj-Rel | Count-Dis | | Event-M | Event-S | |
| *proprietary models* | | | | | | | | | | | |
| Gemini-1.5-pro | 91.89 | 50.00 | 82.93 | 76.85 | **76.00** | 50.00 | **55.32** | **60.84** | 70.59 | 57.50 | 63.51 |
| GPT-4o | **94.59** | **63.33** | 80.49 | **80.56** | 74.00 | 50.00 | 42.55 | 55.94 | **73.53** | 67.50 | **70.27** |
| Claude 3.5 Sonnet | 91.89 | 53.33 | 80.49 | 76.85 | 72.00 | 52.17 | 40.43 | 55.24 | 64.71 | 67.50 | 66.22 |
| *Open-source models for multiple images as input* | | | | | | | | | | | |
| Qwen2-VL | 91.89 | 50.00 | **85.37** | 77.78 | 72.00 | 54.35 | 38.30 | 55.24 | 52.94 | 45.00 | 48.65 |
| Idefics3-8B-Llama3 | 89.19 | 53.33 | **85.37** | 77.78 | 64.00 | 50.00 | 42.55 | 52.45 | 61.76 | 52.50 | 56.76 |
| MiniCPM-V 2.6 | 89.19 | 53.33 | 73.17 | 73.15 | 64.00 | 47.83 | 25.53 | 46.15 | 47.06 | 42.50 | 44.59 |
| mPLUG-Owl3 | 81.08 | 43.33 | 73.17 | 67.59 | 70.00 | 34.78 | 19.15 | 41.96 | 55.88 | 40.00 | 47.30 |
| LLaVA-OneVision | 62.16 | 46.67 | 73.17 | 62.04 | 64.00 | 52.17 | 23.40 | 46.85 | 61.76 | 47.50 | 54.05 |
| InternVL2-8B | 78.38 | 50.00 | 80.49 | 71.30 | 62.00 | 41.30 | 31.91 | 45.45 | 35.29 | 60.00 | 48.65 |
| Mantis | 91.89 | 40.00 | 70.73 | 69.44 | 70.00 | 50.00 | 19.15 | 46.85 | 52.94 | 40.00 | 45.95 |
| Idefics2-8B | 75.68 | 60.00 | 70.73 | 69.44 | 60.00 | 39.13 | 19.15 | 39.86 | 61.76 | 52.50 | 56.76 |
| Brote-IM-XL-3B | 70.27 | 43.33 | 65.85 | 61.11 | 62.00 | 41.30 | 42.55 | 48.95 | 47.06 | 35.00 | 40.54 |
| Idefics2-8B-base | 86.49 | 43.33 | 78.05 | 71.30 | 54.00 | 36.96 | 23.40 | 38.46 | 50.00 | 27.50 | 37.84 |
| Brote-IM-XXL-11B | 70.27 | 40.00 | 65.85 | 60.19 | 56.00 | 47.83 | 31.91 | 45.45 | 55.88 | 32.50 | 43.24 |
| MMICL-XXL-11B | 62.16 | 53.33 | 63.41 | 60.19 | 56.00 | 47.83 | 34.04 | 46.15 | 52.94 | 32.50 | 41.89 |
| MMICL-XL-3B | 32.43 | 50.00 | 65.85 | 50.00 | 52.00 | 45.65 | 38.30 | 45.45 | 41.18 | 32.50 | 36.49 |
| *Open-source models for single image as input* | | | | | | | | | | | |
| MiniCPM-Llama3-V-2.5 | **94.59** | 36.67 | 82.93 | 74.07 | 66.00 | 50.00 | 48.94 | 55.24 | 41.18 | 57.50 | 50.00 |
| GLM-4V-9B | 86.49 | 53.33 | 80.49 | 75.00 | 62.00 | 30.43 | 40.43 | 44.76 | 55.88 | 52.50 | 54.05 |
| InternVL-Vicuna-13B | 62.16 | 46.67 | 60.98 | 57.41 | 64.00 | 54.35 | 25.53 | 48.25 | 58.82 | 60.00 | 59.46 |
| InternVL-Vicuna-7B | 72.97 | 50.00 | 60.98 | 62.04 | 60.00 | 45.65 | 34.04 | 46.85 | 55.88 | 47.50 | 51.35 |
| InternVL-Vicuna-13B-448px | 45.95 | 40.00 | 56.10 | 48.15 | 62.00 | **56.52** | 25.53 | 48.25 | 50.00 | 47.50 | 48.65 |
| mPLUG-Owl2-7B | 64.86 | 40.00 | 53.66 | 53.70 | 60.00 | 47.83 | 19.15 | 42.66 | 52.94 | 50.00 | 51.35 |
| Mini-Gemini-7B-HD | 72.97 | 53.33 | 43.90 | 56.48 | 56.00 | 43.48 | 25.53 | 41.96 | 58.82 | 42.50 | 50.00 |
| SEAL | 56.76 | 43.33 | 53.66 | 51.85 | 54.00 | 41.30 | 19.15 | 38.46 | 29.41 | 40.00 | 35.14 |
| Mini-Gemini-7B | 59.46 | 46.67 | 43.90 | 50.00 | 36.00 | 39.13 | 29.79 | 34.97 | 32.35 | 32.50 | 32.43 |

Table 11: Results on sub-classes of Shifting-E (the easy-level shifting evaluation), where initial views contain clues for answering the question.

| Models | Type I | | | AVG | Type II | | | AVG | Type III | | AVG |
|---|---|---|---|---|---|---|---|---|---|---|---|
| | Geo-Loc | Orient | Daily-Loc | | Obj-Attr | Obj-Rel | Count-Dis | | Event-M | Event-S | |
| *proprietary models* | | | | | | | | | | | |
| Gemini-1.5-pro | 91.89 | 63.33 | **90.24** | **83.33** | 68.00 | 50.00 | **48.94** | **55.94** | **79.41** | 52.50 | 64.86 |
| GPT-4o | **97.30** | 53.33 | 85.37 | 80.56 | 64.00 | 52.17 | 44.68 | 53.85 | 73.53 | **72.50** | **72.97** |
| Claude 3.5 Sonnet | 94.59 | **70.00** | 80.49 | 82.41 | 66.00 | 52.17 | 38.30 | 52.45 | 70.59 | 65.00 | 67.57 |
| *Open-source models for multiple images as input* | | | | | | | | | | | |
| Qwen2-VL | 94.59 | 53.33 | 85.37 | 79.63 | 68.00 | 60.87 | 40.43 | 56.64 | 50.00 | 50.00 | 50.00 |
| Idefics3-8B-Llama3 | 89.19 | 46.67 | 82.93 | 75.00 | 60.00 | 56.52 | 40.43 | 52.45 | 58.82 | 42.50 | 50.00 |
| MiniCPM-V 2.6 | 89.19 | 63.33 | 78.05 | 77.78 | **76.00** | 58.70 | 23.40 | 53.15 | 52.94 | 52.50 | 52.70 |
| mPLUG-Owl3 | 83.78 | 46.67 | 78.05 | 71.30 | 62.00 | 52.17 | 27.66 | 47.55 | 52.94 | 50.00 | 51.35 |
| LLaVA-OneVision | 70.27 | 43.33 | 70.73 | 62.96 | **76.00** | 60.87 | 29.79 | 55.94 | 58.82 | 45.00 | 51.35 |
| InternVL2-8B | 83.78 | 63.33 | 63.41 | 70.37 | 66.00 | 56.52 | 36.17 | 53.15 | 44.12 | 67.50 | 56.76 |
| Mantis | 91.89 | 36.67 | 70.73 | 68.52 | 72.00 | 52.17 | 23.40 | 49.65 | 50.00 | 45.00 | 47.30 |
| Idefics2-8B | 83.78 | 56.67 | 78.05 | 74.07 | 68.00 | 43.48 | 25.53 | 46.15 | 52.94 | 55.00 | 54.05 |
| Brote-IM-XL-3B | 62.16 | 40.00 | 68.29 | 58.33 | 64.00 | 50.00 | 44.68 | 53.15 | 41.18 | 45.00 | 43.24 |
| Idefics2-8B-base | 81.08 | 43.33 | 85.37 | 72.22 | 62.00 | 39.13 | 25.53 | 42.66 | 41.18 | 27.50 | 33.78 |
| Brote-IM-XXL-11B | 64.86 | 33.33 | 60.98 | 54.63 | 58.00 | 52.17 | 46.81 | 52.45 | 41.18 | 42.50 | 41.89 |
| MMICL-XXL-11B | 62.16 | 36.67 | 60.98 | 54.63 | 62.00 | 41.30 | 46.81 | 50.35 | 41.18 | 42.50 | 41.89 |
| MMICL-XL-3B | 56.76 | 46.67 | 68.29 | 58.33 | 56.00 | 45.65 | 40.43 | 47.55 | 41.18 | 35.00 | 37.84 |
| *Open-source models for single image as input* | | | | | | | | | | | |
| MiniCPM-Llama3-V-2.5 | 91.89 | 46.67 | 80.49 | 75.00 | 58.00 | 45.65 | 44.68 | 49.65 | 47.06 | 57.50 | 52.70 |
| GLM-4V-9B | 94.59 | 56.67 | 78.05 | 77.78 | 66.00 | 47.83 | 36.17 | 50.35 | 52.94 | 57.50 | 55.41 |
| InternVL-Vicuna-13B | 59.46 | 43.33 | 65.85 | 57.41 | 60.00 | **63.04** | 36.17 | 49.65 | 52.94 | 52.50 | 52.70 |
| InternVL-Vicuna-7B | 64.86 | 53.33 | 58.54 | 59.26 | 62.00 | 45.65 | 29.79 | 46.15 | 55.88 | 45.00 | 50.00 |
| LLaVA-1.6 13B | 70.27 | 46.67 | 68.29 | 62.96 | 72.00 | 43.48 | 29.79 | 48.95 | 50.00 | 67.50 | 59.46 |
| InternVL-Vicuna-13B-448px | 56.76 | 40.00 | 51.22 | 50.00 | 54.00 | 58.70 | 27.66 | 46.85 | 50.00 | 47.50 | 48.65 |
| mPLUG-Owl2-7B | 67.57 | 43.33 | 56.10 | 56.48 | 50.00 | 47.83 | 23.40 | 40.56 | 47.06 | 47.50 | 47.30 |
| Mini-Gemini-7B-HD | 67.57 | 53.33 | 51.22 | 57.41 | 52.00 | 43.48 | 29.79 | 41.96 | 52.94 | 40.00 | 45.95 |
| SEAL | 56.76 | 43.33 | 53.66 | 51.85 | 52.00 | 36.96 | 25.53 | 38.46 | 29.41 | 45.00 | 37.84 |
| Mini-Gemini-7B | 62.16 | 36.67 | 36.59 | 45.37 | 40.00 | 45.65 | 19.15 | 34.97 | 32.35 | 35.00 | 33.78 |

Table 12: Results on each sub-classes of Shifting-M (the medium-level shifting evaluation), where initial views contain only partial clues for answering the questions.

| Models | Type I | | | AVG | Type II | | | AVG | Type III | | AVG |
|---|---|---|---|---|---|---|---|---|---|---|---|
| | Geo-Loc | Orient | Daily-Loc | | Obj-Attr | Obj-Rel | Count-Dis | | Event-M | Event-S | |
| *proprietary models* | | | | | | | | | | | |
| Gemini-1.5-pro | 89.19 | 60.00 | **90.24** | **81.48** | 70.00 | 47.83 | 42.55 | 53.85 | **73.53** | 60.00 | 66.22 |
| GPT-4o | **94.59** | 53.33 | 85.37 | 79.63 | 64.00 | 52.17 | 42.55 | 53.15 | 67.65 | **70.00** | **68.92** |
| Claude 3.5 Sonnet | 89.19 | 46.67 | 75.61 | 72.22 | 66.00 | 54.35 | 40.43 | 53.85 | 55.88 | 52.50 | 54.05 |
| *Open-source models for multiple images as input* | | | | | | | | | | | |
| Qwen2-VL | 91.89 | 50.00 | 85.37 | 77.78 | **76.00** | 60.87 | 40.43 | **59.44** | 55.88 | 52.50 | 54.05 |
| Idefics3-8B-Llama3 | 86.49 | 50.00 | 80.49 | 74.07 | 64.00 | 52.17 | 40.43 | 52.45 | 55.88 | 50.00 | 52.70 |
| MiniCPM-V 2.6 | 86.49 | **63.33** | 73.17 | 75.00 | 62.00 | 56.52 | 21.28 | 46.85 | 55.88 | 55.00 | 55.41 |
| mPLUG-Owl3 | 81.08 | 46.67 | 68.29 | 66.67 | 62.00 | 54.35 | 25.53 | 47.55 | 58.82 | 52.50 | 55.41 |
| LLaVA-OneVision | 56.76 | 46.67 | 63.41 | 56.48 | 62.00 | 56.52 | 27.66 | 48.95 | 64.71 | 42.50 | 52.70 |
| InternVL2-8B | 78.38 | 50.00 | 63.41 | 64.81 | 66.00 | 43.48 | 31.91 | 47.55 | 44.12 | 50.00 | 47.30 |
| Mantis | 89.19 | 36.67 | 65.85 | 65.74 | 62.00 | 54.35 | 19.15 | 45.45 | 55.88 | 42.50 | 48.65 |
| Idefics2-8B | 67.57 | **63.33** | 70.73 | 67.59 | 56.00 | 43.48 | 21.28 | 40.56 | 52.94 | 50.00 | 51.35 |
| Brote-IM-XL-3B | 48.65 | 36.67 | 63.41 | 50.93 | 54.00 | 50.00 | 44.68 | 49.65 | 44.12 | 35.00 | 39.19 |
| Idefics2-8B-base | 75.68 | 43.33 | 82.93 | 69.44 | 52.00 | 41.30 | 21.28 | 38.46 | 41.18 | 25.00 | 32.43 |
| Brote-IM-XXL-11B | 56.76 | 30.00 | 63.41 | 51.85 | 42.00 | 41.30 | 42.55 | 41.96 | 44.12 | 37.50 | 40.54 |
| MMICL-XXL-11B | 56.76 | 40.00 | 63.41 | 54.63 | 50.00 | 41.30 | 42.55 | 44.76 | 44.12 | 35.00 | 39.19 |
| MMICL-XL-3B | 40.54 | 43.33 | 68.29 | 51.85 | 48.00 | 47.83 | 40.43 | 45.45 | 44.12 | 32.50 | 37.84 |
| *Open-source models for single image as input* | | | | | | | | | | | |
| MiniCPM-Llama3-V-2.5 | 91.89 | 46.67 | 73.17 | 72.22 | 58.00 | 43.48 | **46.81** | 49.65 | 47.06 | 57.50 | 52.70 |
| GLM-4V-9B | 89.19 | 56.67 | 73.17 | 74.07 | 50.00 | 43.48 | 38.30 | 44.06 | 55.88 | 50.00 | 52.70 |
| InternVL-Vicuna-13B | 67.57 | 40.00 | 65.85 | 59.26 | 56.00 | 56.52 | 23.40 | 45.45 | 55.88 | 60.00 | 58.11 |
| InternVL-Vicuna-7B | 62.16 | 43.33 | 63.41 | 57.41 | 62.00 | 45.65 | 25.53 | 44.76 | 52.94 | 52.50 | 52.70 |
| LLaVA-1.6 13B | 62.16 | 53.33 | 65.85 | 61.11 | 62.00 | 50.00 | 29.79 | 47.55 | 52.94 | 50.00 | 51.35 |
| InternVL-Vicuna-13B-448px | 43.24 | 46.67 | 53.66 | 48.15 | 60.00 | 50.00 | 27.66 | 46.15 | 50.00 | 47.50 | 54.05 |
| mPLUG-Owl2-7B | 59.46 | 40.00 | 58.54 | 53.70 | 54.00 | 50.00 | 23.40 | 42.66 | 47.06 | 47.50 | 47.30 |
| Mini-Gemini-7B-HD | 62.16 | 56.67 | 39.02 | 51.85 | 56.00 | 47.83 | 25.53 | 43.36 | 58.82 | 42.50 | 50.00 |
| SEAL | 56.76 | 43.33 | 48.78 | 50.00 | 52.00 | 34.78 | 23.40 | 37.06 | 26.47 | 40.00 | 33.78 |
| Mini-Gemini-7B | 72.97 | 36.67 | 43.90 | 51.85 | 34.00 | 39.13 | 21.28 | 31.47 | 35.29 | 27.50 | 31.08 |

# G  DISCUSSION ON IMAGE SPLITTING AND PROCESSING STRATEGIES

## G.1  IMAGE SPLITTING SETTINGS

In our final pipelines, the original images are equally split into 4 views. We also conduct experiments of splitting into more views and report the results in Table 14. We found that the 4 sub-image setting

Table 13: Results on each sub-classes of Shifting-H (the hard-level shifting evaluation), where initial views do not display clues for answering the questions. Models should decide whether to shift to the next view all by themselves.

| Models | Type I | | | AVG | Type II | | | AVG | Type III | | AVG |
|---|---|---|---|---|---|---|---|---|---|---|---|
| | Geo-Loc | Orient | Daily-Loc | | Obj-Attr | Obj-Rel | Count-Dis | | Event-M | Event-S | |
| *proprietary models* | | | | | | | | | | | |
| Gemini-1.5-pro | **91.89** | **66.67** | **90.24** | **84.26** | 70.00 | 50.00 | **46.81** | 55.94 | 73.53 | 57.50 | 64.86 |
| GPT-4o | **91.89** | 53.33 | 82.93 | 77.78 | 66.00 | 47.83 | 34.04 | 49.65 | **79.41** | **65.00** | **71.62** |
| Claude 3.5 Sonnet | **91.89** | 46.67 | 73.17 | 72.22 | 68.00 | 52.17 | 34.04 | 51.75 | 67.65 | 62.50 | 64.86 |
| *Open-source models for multiple images as input* | | | | | | | | | | | |
| Qwen2-VL | **91.89** | 50.00 | 82.93 | 76.85 | **78.00** | 52.17 | 40.43 | **57.34** | 50.00 | 47.50 | 48.65 |
| Idefics3-8B-Llama3 | 83.78 | 53.33 | 85.37 | 75.93 | 68.00 | 52.17 | 36.17 | 52.45 | 58.82 | 47.50 | 52.70 |
| MiniCPM-V 2.6 | 86.49 | 50.00 | 75.61 | 72.22 | 64.00 | 50.00 | 23.40 | 46.15 | 52.94 | 47.50 | 50.00 |
| mPLUG-Owl3 | 78.38 | 43.33 | 70.73 | 65.74 | 54.00 | 50.00 | 25.53 | 43.36 | 58.82 | 52.50 | 55.41 |
| InternVL2-8B | 64.86 | 40.00 | 65.85 | 58.33 | 62.00 | **54.35** | 34.04 | 50.35 | 52.94 | 40.00 | 45.95 |
| LLaVA-OneVision | 56.76 | 40.00 | 63.41 | 54.63 | 54.00 | 52.17 | 27.66 | 44.76 | 55.88 | 40.00 | 47.30 |
| Mantis | 86.49 | 36.67 | 63.41 | 63.89 | 66.00 | 50.00 | 19.15 | 45.45 | 58.82 | 40.00 | 48.65 |
| Idefics2-8B | 62.16 | **66.67** | 65.85 | 64.81 | 52.00 | 41.30 | 23.40 | 39.16 | 52.94 | 42.50 | 47.30 |
| Brote-IM-XL-3B | 54.05 | 43.33 | 58.54 | 52.78 | 52.00 | 36.96 | 40.43 | 43.36 | 50.00 | 35.00 | 41.89 |
| Idefics2-8B-base | 81.08 | 46.67 | 75.61 | 69.44 | 50.00 | 43.48 | 19.15 | 37.76 | 41.18 | 27.50 | 33.78 |
| Brote-IM-XXL-11B | 56.76 | 30.00 | 60.98 | 50.93 | 44.00 | 34.78 | 38.30 | 39.16 | 50.00 | 35.00 | 41.89 |
| MMICL-XXL-11B | 64.86 | 46.67 | 58.54 | 57.41 | 50.00 | 30.43 | 36.17 | 39.16 | 50.00 | 32.50 | 40.54 |
| MMICL-XL-3B | 43.24 | 43.33 | 68.29 | 52.78 | 46.00 | 32.61 | 38.30 | 39.16 | 52.94 | 32.50 | 41.89 |
| *Open-source models for single image as input* | | | | | | | | | | | |
| MiniCPM-Llama3-V-2.5 | **91.89** | 36.67 | 78.05 | 71.30 | 60.00 | 45.65 | 42.55 | 49.65 | 50.00 | 55.00 | 52.70 |
| GLM-4V-9B | 89.19 | 50.00 | 73.17 | 72.22 | 40.00 | 34.78 | 38.30 | 37.76 | 58.82 | 50.00 | 54.05 |
| InternVL-Vicuna-13B | 56.76 | 36.67 | 60.98 | 52.78 | 62.00 | 50.00 | 21.28 | 44.76 | 61.76 | 60.00 | 60.81 |
| InternVL-Vicuna-7B | 59.46 | 43.33 | 63.41 | 56.48 | 52.00 | 45.65 | 23.40 | 40.56 | 55.88 | 50.00 | 52.70 |
| LLaVA-1.6 13B | 51.35 | 50.00 | 60.98 | 54.63 | 58.00 | 41.30 | 29.79 | 43.36 | 55.88 | 52.50 | 54.05 |
| InternVL-Vicuna-13B-448px | 48.65 | 40.00 | 60.98 | 50.93 | 58.00 | 47.83 | 29.79 | 45.45 | 50.00 | 52.50 | 51.35 |
| mPLUG-Owl2-7B | 56.76 | 40.00 | 56.10 | 51.85 | 58.00 | 45.65 | 25.53 | 43.36 | 52.94 | 42.50 | 47.30 |
| Mini-Gemini-7B-HD | 70.27 | 50.00 | 51.22 | 57.41 | 52.00 | 47.83 | 29.79 | 43.36 | 58.82 | 47.50 | 52.70 |
| SEAL | 56.76 | 36.67 | 51.22 | 49.07 | 54.00 | 34.78 | 23.40 | 37.76 | 29.41 | 37.50 | 33.78 |
| Mini-Gemini-7B | 64.86 | 43.33 | 51.22 | 53.70 | 36.00 | 28.26 | 23.40 | 29.37 | 26.47 | 20.00 | 22.97 |

is able to derive fair and reliable evaluation results, which is not only effective but also efficient. More splits require additional inference time and resources (e.g., the context length, GPU memory, etc.), but they only yield similar trends and conclusions compared to 4 sub-image setting.

Additionally, there are two issues with more splits. First, it is challenging for the ability to process multiple images and understand their relationships. As shown in the table above, when increasing the number of splits, LLaVA-1.6-7b degrades from 60.31 to 57.69 (-2.62) on average, and LLaVA-1.6-13b decreases 1.27 on average. Although increasing the splits would increase the performance of zooming evaluation, the performance of shifting is remarkably decreased. As we focus on active perception concerning both zooming and shifting, a split of 4 would present a decent balance. Second, the necessary information would be more likely to be split into different tiles, causing information loss.

Table 14: Experimental results of different splits.

| Model | Splits | Zooming | Shifting-R | AVG |
|---|---|---|---|---|
| LLaVA-1.6 7B | 4 | 68.92 | 51.69 | 60.31 |
| LLaVA-1.6 7B | 6 | 73.23 | 53.85 | 63.54 |
| LLaVA-1.6 7B | 8 | 72.92 | 48.61 | 60.77 |
| LLaVA-1.6 7B | 9 | 66.46 | 46.16 | 56.31 |
| LLaVA-1.6 7B | 16 | 69.23 | 46.15 | 57.69 |
| LLaVA-1.6 13B | 4 | 65.23 | 53.85 | 59.54 |
| LLaVA-1.6 13B | 6 | 71.69 | 46.46 | 59.07 |
| LLaVA-1.6 13B | 8 | 71.84 | 44.00 | 57.92 |
| LLaVA-1.6 13B | 9 | 72.00 | 43.69 | 57.84 |
| LLaVA-1.6 13B | 16 | 73.31 | 43.23 | 58.27 |

## G.2 Strategies of Processing Multiple Images for Single-image Models

For all the pipelines, multiple views might be selected depending on the response of models, which can be naturally handled by multi-image models. However, for models that only accepts single image per input, we apply different image processing approaches for zooming and shifting pipelines. For the shifting pipeline, we proposed to concatenate the selected views or convert them into textual descriptions to fit the information of multiple images into a single input. The concatenation refer to stitch the images selected views together from left to right to form a single image as the input for the model. This is applicable for both missing view examination stage and question answering stage. For the question answering stage in zooming pipeline, if multiple views were selected in the first stage of our pipelines, we will use the each selected view to ask questions sequentially. After obtaining answers, if the model answers correctly based on any of the views, we consider it a complete and successful view selection.

Table 15: Experimental results providing single-image models with captions as compensation for the invisibility of previous images.

| Model | Visual Info. Type | Zooming | Shifting-R |
|---|---|---|---|
| LLaVA-1.6 7B | Image concatenation | 68.92 | 51.69 |
| | Textual descriptions | 60.31 -8.61 | 53.83 +2.14 |
| LLaVA-1.6 13B | Image concatenation | 65.23 | 45.85 |
| | Textual descriptions | 60.00 -5.23 | 43.69 -2.16 |
| mPLUG-Owl2 7B | Image concatenation | 55.38 | 47.38 |
| | Textual descriptions | 62.77 +7.39 | 54.15 +6.77 |
| MiniCPM-Llama3-V-2.5 | Image concatenation | 61.25 | 60.92 |
| | Textual descriptions | 61.25 -0 | 60.31 -0.61 |

In addition to the directly processing of image, we also propose methods to deliver visual information by converting images into textual descriptions. This enables single-image models to "see" multiple images in the form of text inputs. This method can be applied to both shifting and zooming settings. When multiple views are required, we preserve merely the current view in the form of image, while converting the remainings into textual descriptions via the prompt "Please describe the image:". Results of typical single-image models, LLaVA-1.6, mPLUG-Owl2 and MiniCPM-Llama3-V-2.5 are shown in Table 15.

For the strategy of converting image into text, it is supposed to be a compensation for the image concatenation strategy to avoid images being resized. On the contrary, we observe significant drops of results on both zooming and shifting evaluations for most of the investigated models, indicating that the resizing issue of image concatenation strategy has minor influence on the performance. Moreover, the operation to converting images into textual descriptions introduces the influence of other abilities that interferes the evaluation of active perception abilities.

## H Case Study of Providing Human-annotated Clues

We present a case study of ActiView in Figure 6. The first question targets at the most expensive watermelon, and only two out of four price tags, the "39" and "59" ones, are standing for the prices of watermelons. A distracting information appears at the "69" price tag that corresponds to papayas instead of watermelons. Models easily mislead by the most expensive tag "69" during evaluation. However, when we provide the models with the view of the price tags and remind them to focus on these tags, both GPT-4o and GPT-4V models correctly answer the question, indicating that actively perceiving key information helps improve model performance. While Gemini-1.5-pro gives the correct answer both with and without hints, and Gemini-1.5-pro fails to benefit from the hints. The second question asks models to recognize the place of the picture. Although it may be difficult to distinguish at first glance, we can still identify this place as an airport from some details, such as a airline's logo. Since there isn't a need to extract much information from the image, and there is little distracting information, all the four models answered the question correctly both with and without hints.

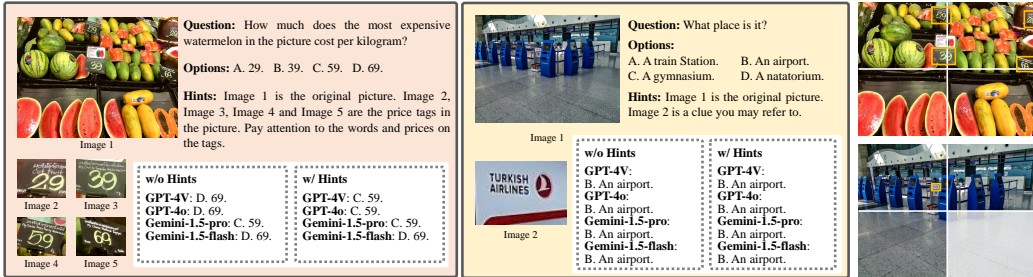

Figure 6: Two cases of ActiView benchmark when given human-annotated visual clues for shifting and zooming evaluation. Left: The questions and answers of models. Right: We show the location of the visual clues we provided in the original image, as well as the areas chosen by GPT-4o model. For the first case, GPT-4o chooses all the areas, and for the second case, it chooses all the areas except the one in the bottom right corner.

The right side of Figure 6 shows a comparison between the attention areas selected autonomously by GPT-4o and the areas highlighted by the hints we provided. It can be observed that when facing some difficult problems, although the model selects all the regions, it is unable to actively retrieve all the necessary details, thus lacking some essential information for answering the question. When the questions are relatively simple, the model successfully identify important information and gives the correct answer. This indicates that the GPT-4o model possesses a limited level of active perception capability and it still has room for improvement. We have also observed similar conclusions for other models.

## I  LIMITATIONS AND FUTURE TOPICS

In this study, we utilize a specific form of Visual Question Answering (VQA) to assess active perception abilities of models. While this form of VQA presents significant challenges for current multimodal language models (MLLMs), it does not encompass all aspects of active perception. For instance, it overlooks factors such as perspective distortion, multi-sensor integration, and the incorporation of more dynamic or interactive environments. Moreover, this study solely evaluates the inherent capabilities of the MLLMs. Techniques like tool learning and multi-agent collaboration could potentially enhance active perception performance based on existing MLLMs, making these areas worthy for future exploration and improvement. Given that these limitations exceed the scope of a single paper, we will leave them as future work.

## J    Prompt Template

In this section, we will provide detailed templates used for evaluation pipelines depicted in Figure 4.

### J.1    Templates for General Question Answering

The general VQA template that requires models to answer questions given images is as following:

---

**An Example Prompt for General Question Answering**

```
Carefully analysis this image <image>, and answer the question from the given
options. Question: <question> Options: <options>. Answer:
```

---

We develop a different template for two of our evaluated models, SEAL and MGM series. These models are optimized especially on VQA tasks, and sometimes fail to strictly following long textual instructions. Therefore, we use a simple and straightforward template to prompt these models for answers as follows:

---

**An Example Prompt for Question Answering(SEAL and MGM)**

```
<question> <options>. Answer:<image>
```

---

### J.2    Templates for Zooming Evaluation

Here are templates used in the two stages of zooming pipeline depicted in Figure 4 (a). Note that the term "description_of_splits" refers to the positions of the views that guide the model to shift and select views. "description_of_splits" varies depending on how the views are divided. Taking 4 sub-image for example, it is described as "1 is the upper-left part, 2 is the lower-left part, 3 is the upper-right part, and 4 is the lower-right part." The model should then response with "1, 2, 3, and/or 4" to select the appropriate views. The prompts are as follows:

---

**An Example Prompt for View Selection**

```
This is the full image <image>, which is split in to <num_splits> equal parts,
numbered from 1 to <num_splits>, where <description_of_splits>.
===
Response with the number of part (at least one part, at most <num_splits> parts),
that must be used to answer the question. The question is: <question>
===
Do not directly answer the given question. Response with the selected number of
parts, split by ' if there are multiple selections. Your Response:
```

---

**An Example Prompt for Zooming Question Answering**

```
Image 0 is the full image. <zoomed_images> These are your selected part from the
full image to be zoomed for details for answering the question. Please answer
question according to the given images from the the given options. Question:
<question> Options: <option>. Answer:
```

---

### J.3    Templates for Shifting Evaluation

Here are templates used in the two stages of zooming pipeline depicted in Figure 4 (b).

**An Example Prompt for Missing-view Examination**

```
You will be presented with a partial image and a question concerning the full
image. image 0 is <image0>, is the <image_view> part of the full image. Given image
0, please determine if you need more visual information to answer the question:
<question>
===
Do not directly answer the question. If you can answer the question without more
visual information, response with NO. Otherwise, response with other image parts
you need to see given this <image_view> part, you can choose from these views:
<view_options>. Your Response:
```

**An Example Prompt for Shifting Question Answering**

```
These are parts of an image. <all_required_views>. Carefully analysis these images
and pay attention to their original position. Answer the question from the given
options. Question: <question>. Options: <option>. Answer:
```

## J.4 TEMPLATES FOR MIXED EVALUATION

Here are templates used for the mixed pipeline depicted in Figure 4 (c). We design two templates for regarding the type of current view. We apply template "Operation Determination"(1) from the followings for the full images, and apply template "Operation Determination"(2) from the followings for zoomed views. Templates are as follows:

**An Example Prompt for Operation Determination (1)**

```
You will be presented with a full image <image> and a corresponding question to
answer. The image is split in to <num_splits> equal parts, numbered from 1 to
<num_splits>, where <description_of_splits>.
You can check for detailed visual information via zooming operation that zoom in
to your selected part or parts iwth from the above numbers. Response with the the
numbers of parts you wish to zoom in, or response with ''none'' if you don't need
to can check for details.
The quesiton is: <question>
You should not directly answer the question. You should generate the a json dict
containing 2 fields:
- ''part'': type str, the selected numbers of index of parts, split by '','', or
'none' if no zooming required;
- ''reason'': type str, why you choose these parts.
Your response:
```

---

**An Example Prompt for Operation Determination (2)**

```
You will be presented with a partial image and a question concerning the full
image. image 0 is <image0>, is the <image_view> part of the full image. Given image
0, please determine if you need more visual information to answer the question:
<question>
===
Your are given a full image <image> and a corresponding question to answer. The
image is split in to <num_splits> equal parts, numbered from 1 to <num_splits>, where
<description_of_splits>. Your have chosen to zoom in to these parts, <zoomed_images>,
for detailed checking if they can help to ansewr the quesiton.
Question: <question> Options: <option>.
Now, there are two operations: ''keep'' and ''shift''.
- ''keep'': choose none or more parts from the zoomed ones to answer the question;
- ''shift'': you can shift to the rest parts to answer questions or answer question
with none sub-parts.
You should not directly answer the question. You should return you answer in a json
dict containing two fields:
- ''zoom_keep'': type str, the index numbers of required parts split by '','', or
''none'' if the zoomed parts are useless;
- ''shift'': type str, the index numbers of the rest parts, that are useful to the
question split by '','', or ''none'' if you don't wish to shift.
Your response:
```

---

**An Example Prompt of Quesiton Ansewring for Mixed Pipeline**

```
Image 0 is the full image. <image_views> <image_view_desc> These are your selected
part of image that must be used to answer the question. Please answer question
according to the given images from the the given options. Question: <question>
Options: <option>. Answer:
```

## K    ATTEMPTS OF AUTOMATIC DATA GENERATION

In this last section, we discuss our experiments of automatic data generation, and analyse why powerful models like GPT-4V fail to accomplish this task. We will discuss the process and demonstrate typical failure cases in the following sections.

### K.1    AUTOMATIC DATA GENERATION PROCESS

In the process of automatic data generation, we used the GPT-4V model for the following experiments:

- **Step 1**: We applied heuristic prompts on public datasets to encourage GPT to generate creative annotations across all types.

- **Step 2**: We selected the types that showed the best performance in automatic annotation and conducted batch annotation specifically for these types.

- **Step 3**: We manually filtered a subset of data that could be used.

In **Step 1**, we not only employed heuristic prompts to encourage GPT to generate diverse annotations but also specified the annotation types and their precise meanings (provided as candidates, encouraging the model to select from them). We restricted the annotation fields and types, and provided several manually curated examples as few-shot instances. Considering that some images in public datasets may not be suitable for our task, we allowed GPT to return "None" for images deemed unsuitable for annotation. The filtered annotation data were then re-evaluated using a scoring prompt, where we provided our annotation types and requirements, instructing GPT to rank the annotated data to assess its suitability.

In **Step 2**, we found that GPT performed best in annotating data of the counting type (based on a combination of manual inspection of the annotation results and GPT's automatic scoring). There-

fore, we decided to use GPT for automatic annotation of counting-type data. Considering that some public datasets (such as VCR) contain images with more than one type of bounding box, we processed different bounding box types in batches for each image to ensure that only one type of object was counted at a time.

Detailed prompt templates are attached in the third sub-section of this section.

## K.2 CASES OF UNSUCCESSFUL GENERATIONS OF GPT-4V

We provide two typical cases demonstrating why GPT-4V fail to generate usable instances. The corresponding image is Figure 7. For the case regarding the left image, it presents a typical encountered issue case of hallucination and speculation without a factual basis. Given this image, GPT-4V produces the following annotations prompted by **Step 1**:

```
{''question'': ''Which of the following best describes the setting based on the
appearance and arrangement of the glass items on the table?'',
''options'': [''A casual family dinner'', ''A quick lunch at a fast food restaurant'',
''An official or formal meeting'', ''An outdoor picnic''],
''answer'': 2,
''groundtruth'': ''The setting seems to be an official or formal meeting given
the presence of multiple large, elegant glasses on the table, which suggest formal
drinkware typically used in such settings.''}
```

The question and annotated answer posed by GPT-4V makes certain assumptions about the image that this scenario shows "An official or formal meeting". The question is not answerable concerning only this image, where it could refer to either a meeting or a dinner. Moreover, the other options except for annotated answer does not match the image in any circumstances, and can be easily eliminated without any further observation of the image. The answers does not strictly follow the given ground truth (i.e., the answer to the question cannot be rigorously inferred from the visual clues in the image), where the glasses do not support the reasoning. For the case of the right image, it presents a typical failure case from **Step 2**. Regarding this image, GPT-4V generates an ambiguous question "How many umbrellas can be seen in the image?", where there are some small visible objects could potentially be umbrellas as well.

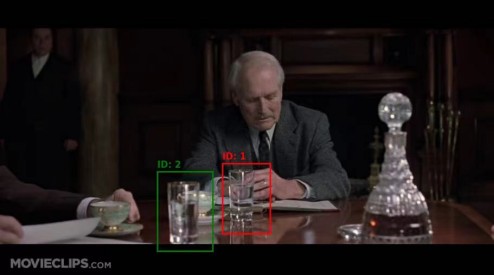 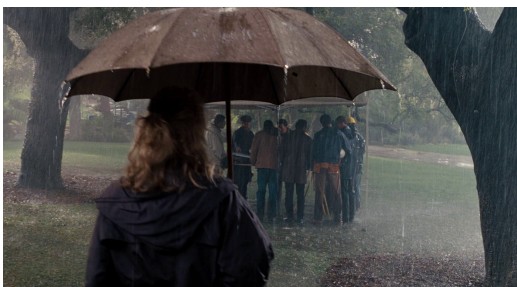

Figure 7: Left: example of the automatic annotation results from Step 1. Right: example of the automatic annotation results from Step 2, where the question annotated by GPT is "How many umbrellas can be seen in the image?"

## K.3 PROMPT TEMPLATES USED FOR AUTOMATIC GENERATION

Here, we give the prompt used for automatic annotation in **Step 1** and **Step 2**.

---

Heuristic prompt used for automatic annotation in **Step 1**

---

The clues for marking information in several bboxes in this picture are: {clues}
Based on several bboxes and corresponding clues, please design a question that
requires the model to synthesize the information in these bboxes (at least two,
and can only be answered based on the information in the bboxes and the clues
corresponding to the annotated information). You only need to ask the question,
and there is no need to repeat the clue again.
Note that the existence of bbox (including its ID information) cannot be mentioned
in the question. Questions and reasoning should be based on objective facts as
much as possible instead of subjective guessing.But at the same time, you should
also avoid grounding questions and questions that can be answered without pictures
(including questions like what someone in the picture is doing)
Next, mark me the corresponding information in the following format:
1. ''question'' (str)
2. ''options'' (list)
3. ''abilities'' (list): choose from ''zoom in'', ''zoom out'', ''shifting'' (it
is mentioned in the analysis and is not mentioned at the beginning)
4. ''answer'' (int, index of option)
5. ''order'': the order in which the pictures cut out of the bbox and the entire
picture are displayed (the list is given in the order of reasoning, all of which
are ints, representing the id corresponding to the bbox on the picture, if it is a
complete picture, it is 0)
6. \groundtruth": Give the reasons and complete reasoning process for answering
the question
7. ''number_of_operations'': For example, first zoom in and then move the angle of
view, it is two operations
You must give me the answer in the following json-string format(not code block) and
dont say anything else:
{{
''question'': question(str),
''options'': options(list),
''abilities'': ablities(list),
''answer'': answer_index(int),
''order'': order(list),
''groundtruth'': groundtruth(str),
''number_of_operations'': number of the operations(int)

---

**Scoring prompt used for automatic annotation in Step 1**

We want to design a question about the picture to test the active perception
ability of the respondent. Here are the requirements:
You will be provided with an image and information of bboxes in it. You should
design a question that requires the respondent to synthesize the information in
these bboxes.
While designing the questions, you must follow these rules:
- The question should be based on the information in the given bboxes.
- The question requires the respondent to obtain information from the field of
view of these bboxes as a basis, identify irrelevant information on the picture,
and move the field of view of different bboxes to obtain more information before
answering the question.
- Differences between options should be distinct. And options must not be conflict
to each other.
- There should be one and only one correct answer among all options.
- The evidence or clues for answering the question must be visible in the image.
Also, you should realize the following conditions:
- The answers must not require the respondent guess subjectively.
- You cannot generate questions require simple object grounding, e.g., what is the
object in a certain region, what is the color of an object, etc.
- The existence of bbox and visual clues (including their ID information) cannot
be mentioned in the question nor in the options.

You should score the annotation through the rules given above.  Here are the
predefined levels for scoring, where level D is the worst and level A is the best:
- Level D: no reasonable questions can be generated for the given image by strictly
following our rules.
- Level C: the question contains subjective guesses and judgments, rather than
strictly following the rules(e.g.  infer the location from the architectural
style/image style rather than some grounding signs and texts etc.)
- Level B: the question can be answered via simple captioning of the pictures(like
using ViT or OCR to caption the picture and ask the language model to answer the
question with out the picture), or can be answered via pure common sense reasoning.
- Level A: the question is cleverly designed and is completely based on the
information in the picture. It requires the respondents to visit different parts
marked on the image for comprehensive reasoning, which fully complies with the
above marking rules.

Remenber, if any subjective guess seems to appear, or anything that requires
inferring from knowledge outside the image, or anything that does not follow our
rule strictly (including asking for some weired questions etc.), do not hesitate
to assign a low level.

Here's the annotation information of the given picture:
{annotation}

You must give me the answer in the following json-string format(not code block) and
don't say anything else:
{{
''score'': string, choose from ''A'', ''B'',''C'', ''D'',
''reason'': string, explain why you give this score

---

Prompt used for automatic annotation in **Step 2**

```
You are an annotator to design questions and options for given images. Here are
the guidebook for you:
===
Overall task description: You will be presented with an image, please generate
a question, corresponding options and answer to the question, and some other
information that help the reasoning process as well.
===
Detailed requirements you **must** follow:
 - You must design the problem in the following type:
 Counting with restricted information or extending reasoning based on counting. For
example, there are lots of products in the image, but only a part of them are on
sale, you can ask for the number of on sale products. Options are list of numbers.
Candidates:
 - How many people are wearing black hat?
 - How many products are on discount?
 - Which color of umbrellas are the most numerous in the picture?
 But remember, you *cannot* ask common sense questions like how many objects are
there in the picture, which can be answered without reasoning.
 - **Simple grounding questions are NOT allowed**, such as (but not restricted to):
``what is xxx object?'', ``What is the color/style of xxx?'', and etc.
 - For answers:
 - By referring to the image, there must exists one and only one answer, without
any ambiguities and subjective guesses.
 - The evidence for answering the question must be visible in the image.
 - Objective reasoning are not allowed.
 - DO NOT rely on information that does not exist in the image.
 - For options:
 - The differences between generated options should be distinct.
 - There should be one and only one correct answer among all options.
 - Options must not be conflict to each other.
===
The requirements of the generated data format are as follows:
1. ``question'' (str, start with wh words or prep + wh words)
2. ``options'' (list)
3. ``abilities'' (list): choose from ``zoom in'', ``zoom out'', ``shift''
4. ``answer'' (int, index of correction option, starting from 0)
5. \groundtruth": Give the reasons and complete reasoning process for answering
the question
6. ``operations'': For example, first zoom in to a region and then moving to a
different region, counted as two operations
===
Here are some bounding boxes and their type for you to refer to:
{boxes}
The items in these bounding boxes are all {type} The questions you ask must be
about the information within the bounding boxes and strictly meet the requirements
and question types given to you above.
===
If it is impossible to come up with required questions, you should response with
``question'': (str)``None'' in json-string format(not code block). Otherwise, you
must generate response in the following json-string format(not code block) and dont́
say anything else:
{{
``question'': question(str),
``options'': options(list),
``abilities'': ablities(list),
``answer'': answer_index(int),
``order'': order(list),
``groundtruth'': groundtruth(str),
``operations'': number of the operations(int),
}}
===
Please generate response for the given image that **strictly follow** the above
requirments:
```

