# OpenReview forum: "ActiView: Evaluating Active Perception Ability for Multimodal Large Language Models"
_ICLR.cc/2025/Conference — ICLR 2025 Conference Withdrawn Submission_

### Official Review · Reviewer_GR23 · 2024-11-03

**Soundness:** 2
**Presentation:** 3
**Contribution:** 2
**Rating:** 3
**Confidence:** 4

**Summary:**

This paper proposes a benchmark, named ActiView, to evaluate active perception in Multimodal Large Language Models (MLLMs). The active perception is evaluated from two aspects: zooming and shifting. They observe that the ability to read and comprehend multiple images simultaneously plays a significant role in enabling active perception

**Strengths:**

1. Evaluating the active perception capabilities of MLLMs is crucial for their real-world applications, and this aspect has not been sufficiently addressed in previous benchmarks.

2. The authors have carefully annotated their dataset, organizing it into eight sub-categories to enable a thorough and nuanced evaluation of MLLMs.

3. The experimental analysis is comprehensive and clearly presented.

**Weaknesses:**

1. The scale of this dataset is too small, only with 314 images, as shown in Table 1.

2. The dataset primarily focuses on categories related to object relationships, attributes, and locations. Previous datasets have been released that cover areas such as object detection, counting, image event extraction, and image hallucination detection. While this dataset addresses active perception, the categories and types of questions it includes have already been covered in earlier datasets, resulting in a lack of novelty.

3. Where is InstructBLIP? Its Q-former can extract instruction-related visual features, playing an active perception role.

**Questions:**

Instead of evaluating the active perception of image content by MLLMs using shifting and zooming, can you evaluate it in other ways?

---

> ### Author Response · Authors · 2024-11-24
> **Response to weaknesses and questions**
>
> Please find our responses as follows:
>
> **Response 1: The scale is too small (corresponding to weakness #1).**
>
> - The smaller size is a trade-off for the high-quality manual annotations. Each image’s integration with shifting and zooming tasks adds depth to the analysis, compensating for the scale .
>
> - We contain 314 unique images and 325 unique questions. We further split the images and disrupt the order to enable shifting and zooming tasks. Altogher, 1625 evaluation instances are generated, which is not a small number.
>
> - Also note that FISBe [1] contains 101 images, V* Bench [2] contains 191 images with manual annotated quesitons, ViP-Bench [3] contains 303 images, and [4] 310 instances.
>
> [1] Mais, L., Hirsch, P., Managan, C., et al. FISBe: A real-world benchmark dataset for instance segmentation of long-range thin filamentous structures. CVPR 2024.
>
> [2] Wu, P., & Xie, S. V*: Guided Visual Search as a Core Mechanism in Multimodal LLMs. CVPR 2024.
>
> [3] Cai, M., Liu, H., Mustikovela, S. K., et al. ViP-LLaVA: Making Large Multimodal Models Understand Arbitrary Visual Prompts. CVPR 2024.
>
> [4] Tian, S., Finn, C., Wu, J., A Control-Centric Benchmark for Video Prediction. ICLR 2023
>
> **Response 2: Categories and novelty (corresponding to weakness #2)**
>
> - ActiView is complementary to datasets like object detection and event-centric datasets. It uniquely challenges models to operate under perceptual constraints, such as shifting and zooming, which these datasets lack.
>
> - It is inappropriate to dismiss ActiView’s novelty solely based on the overlap in task categories with existing datasets. While some categories like object relationships and attributes are common, ActiView is unique in its integration of active perception tasks through zooming and shifting. This dynamic evaluation mechanism is not addressed by previous benchmarks, which primarily rely on static perceptual fields. Moreover, ActiView’s carefully designed pipelines and constrained perceptual fields push models to perform reasoning under limited views—replicating real-world scenarios that demand active exploration. This emphasis on active perception and reasoning processes differentiates ActiView as a novel and specialized benchmark.
>
> - Our benchmark shows that current models do not demonstrate practical active perception ability, and aims a few years ahead of current SOTA.
>
> **Response 3: InstructBLIP (corresponding to weakness #3)**
>
> - As InstructBLIP has been available for over a year, many improved vairent based on this model have been released, driven by the rapid advancements in Artificial Intelligence research. Although InstructBLIP is a very promising model, it lacks the ability to read multiple images simultaneously, and the instruction-following ability also needs improving compared to recent models.
>
> - We evaluate two powerfull varients Brote and MMICL (as we already stated in our paper) that are **derived from InstructBLIP** (as mention in Line 318). Results of these two models are reported in Table 2, 7, 8, 9, 10, 11, and 12. It is out-dated and improfessional to stick to the past even if numbers of advanced approaches are newly presented, and researcher are no longer including simple MLPs and CNNs as baselines for visual-language tasks.
>
> **Response 4: Other evaluations except zooming and shifting**
>
> - Firstly, our paper’s focus on zooming and shifting stems from their clear representation of active perception. Secondly, other methods, such as evaluating temporal dynamics or integrating real-time feedback loops, could be explored in future iterations.
>
> - Plus, we are now working on more dynamic evaluations including segmenting regions semantically. This pipeline will be released when more models (not limited to GPT-4o) are able to follow the specific instuctions.

---

### Official Review · Reviewer_5Vjd · 2024-11-03

**Soundness:** 2
**Presentation:** 3
**Contribution:** 2
**Rating:** 3
**Confidence:** 4

**Summary:**

The paper proposes a new benchmark for multimodal large language models (MLLMs) that focuses on their active perception capabilities. To answer a fine-grained question about the image, models are presented either with a global image view and need to zoom into a specific part of the image, or with a local view that does not necessarily include the required information requiring the model to shift to over adjacent views looking for the correct information. Several closed and open models, 27 in total, are evaluated on this new benchmark.

**Strengths:**

- Active perception is an important research topic that is required for build autonomous agents that can search and find relevant information in our world.
- The paper evaluates a large amount of MLLMs of different kinds, making it a very comprehensive study
- The proposed dataset is the first of a kind that requires both shifting and zooming in an active perception setting.

**Weaknesses:**

- All three pipelines are quite similar to each other. Zooming allows access to the full image and proper selection by the model, while shifting restrict access to the global view and a gradual access to image parts without active selection. Lastly, the mixed pipeline is very similar to the zooming setting, also giving access to the global image view. It is not clear why the mixed pipeline requires the model to discard images. Is it to make it artificially distinct or more difficult? Given that the paper highlights both settings as a key novelty of the provided benchmark (previous benchmark only evaluated one of the two), this small distinction between the pipelines diminishes the value of this contribution.
- The shifting pipeline does not give the model agency over which views it wants to request. Instead, it uses a pre-defined order for the views, i.e., the model just decides "next view" or "stop", without any penalty of requesting more views. The optimal policy is likely to simply request all views to have all available information. One ablation should include "All views" for the Shifting pipeline.
- The experimental results in Table 2 reveal weaknesses of the proposed benchmark:
  * There is not much difference in the zooming pipeline when comparing full image vs. zooming. A significant amount of models even perform better with the full image. This result suggests that zooming is not really necessary for the proposed benchmark.
  * The same is true for a smaller subset of models for the shifting pipeline where single view is better than one or multiple Shifting settings. As acknowledged in the results discussion, the experimental results also do not always follow the difficulty scaling of the setting, e.g., Gemini-1.5-pro achieves its best performance on Shifting-H.
  * These observations raise concerns about one of the main claims of the paper, i.e., that the images-question pairs from the proposed benchmark truly require active perception to be solved.
- The results in Tab. 4 tell a similar story to the concerns of Tab. 2. For multi-image models, all models except for InternVL2-8B achieve close to the GT Acc. on Zooming regardless of their Recall with GPT-4o showing the second highest drop in performance despite having the highest Recall. This suggests that Recall does not have a significant impact on the performance. Similarly for Shifting-R, Gemini-1.5-pro has a low Recall and a 5% drop in performance, while mPLUG-Owl3 has a high Recall and a 9% drop is performance (overall trends are a bit harder to spot because of the ordering).
- L. 475 states a hypothesis that the order of input views helps models in the hard settings as relevant images are more recent. However, if this is true, this is a design flaw of the setting which is supposed to be harder than the others. One option could have been to restrict the number of views a model can request, but it relates to an earlier point that the model cannot actually choose which view to request.
- Tab. 13 shows that adding more views hardly makes a difference which is counter-intuitive. It also suggests that either the image-question pairs or the setting do not really requiring active perception. Having more views and limiting the number of views or accounting for it in the evaluation metric could improve this.
- Case (b) in Fig. 5 showcases the problems above: The models shift through all views with the last one containing the relevant information, then make a wrong decision. From an active perception point of view, the models did everything correctly. Hence, this negative result did not measure their active perception capabilities.

**Questions:**

Please address the points raised in the Weaknesses section. Here are additional relevant questions:
- L. 269 explains that the dataset consists of 314 images with 325 questions, but Tab. 1 denotes 1,625 instances. Could you please explain this discrepancy? What is counted as an instance?
- The Shifting-R setting randomly selects an initial view. Is it randomly selected once for each image and the same initial view given to each model, or does each model receive a different randomly selected initial view?
- How is the adjacent view selected for the Shifting tasks? Is the same order given to all models?
- L. 420 states: "We obtain the random choice result of 33.95%, averaged over 10k runs." How was this random choice calculated? Why does it require making multiple runs? Another important baseline is an LLM (text-only) evaluation to measure the amount of common sense answers.
- What exactly is meant by the "Single View" column in Tab. 2? Which view is given for this experiment?
- L. 460 states that redundant information might distract a model. Why could this be the case? If the model is presented with the right information multiple times it should rather reinforce the correct answer. Can you provide an example? An ablation where all 4 views is given to the model could help better understand this issue.

---

> ### Author Response · Authors · 2024-11-24
> **Response (Part 1)**
>
> Because of the 5000 characters limitation per reply, our responses will be split into 4 parts.
>
> Thanks for your insightful comments, please find our responses and additional experiments as follows:
>
> Response 1-4 correspond to detailed questions:
>
> **Response 1: The dataset statistic (corresponding to quesiton #1)**
>
> - The dataset contains 314 images paired with 325 questions, where some images correspond to more than one question. Each image-question pairs is evaluated across five different initial settings, including one for zooming and mixed pipelines (325 $\times$ 1) and four for shifting pipeline (325 $\times$ 4, varying difficulty levels of the inital view, e.g. random/easy/medium/hard), resulting in a total of 1,625 (i.e. 325 $\times$ 5) evaluation instances. An instance is defined as a single “question-initial view” pair.
>
> **Response 2: The settings of views of shifting pipeline (corresponding to quesiton #2, #3)**
>
> - **Question #2**: Is it randomly selected once for each image and the same initial view given to each model?
> - **Response**: Yes, in the Shifting-R setting, a random initial view is selected once per image, it is selected according to a fixed random seed. Thus, the same initial view is given to all models to ensure consistency in evaluation.
>
> - **Question #3**: How is the adjacent view selected for the Shifting tasks? Is the same order given to all models?
> - **Response**: For the Shifting tasks, adjacent views are selected following a fixed predefined sequential order (upperleft-lowerleft-upperright-lowerright, as mentioned in Appendix I), and it is the same order given to all models.
>
> **Response 3: The random choice results in Table 2 and textonly evaluation (corresponding to quesiton #4)**
>
> - The “random choice result of 33.95%” is calculated by randomly selecting an answer from the available options for each question, averaged over 10,000 runs to minimize noise from random fluctuations. As this is to randomly generate an answer from the options, no input content is provided. Answering according to the images and questions cannot be regarded as RANDOM. This could also be viewed as the distribution of answers. Our options ranges from 2 to 7 (not always 4 options).
>
> - Thanks for your constructive advise. We employ the prompt template for text-only evaluation as following: “Please answer questions based on you commonsense knowledge. If you are not able to answer, please select a most probable one. {Question} {Options} Your answer:” We found that there are some questions that can be answered via commonsense Here are the resutls of text-only evaluation:
>   | Model | Claude | GPT-4o | Qwen2-VL |MiniCPM-V 2.6| Idefics3 |Brote-IM-XL|
>   |:------|:------:|:------:|:--------:|:--------:|:-----------:|:--------:|
>   |  ACC  |  2.14  |  2.45  |  23.38   |    26.77    |  44.92   |  40.00   |
>   |ACC(guess)|26.07| 37.73  |  42.77   |    41.54    |  47.38   |  40.00   |
>
>   - We implement two settings: 1. asking models to answer according to **commonsense ONLY** and reply “None” for questions cannot be answered via commonsense (results shown in Row ACC), and 2. permitting models to guess according to commonsense (results shown in Row ACC(guess)).
>   - The above results (Row ACC) indicate that **questions in our benchmark cannot be simply answered via commonsense**, where two powerful models GPT-4o and Claude achieves only 2.45% and 2.14%.
>   - The row of ACC(guess) presents results of generating the most probable answers. These reflect the bias learnt from the training corpus.
>   - The differences between these two results demonstrate the ability of instruction-following. We found that Idefics3 and Brote-IM-XL present weaker instruction-following ability compared to other models in this table, that they still exhibit a behavior of guessing when commonsense cannot be used to answer the questions.
>   - The corresponding experiments will present in the Appendix of our paper (as there is no available space in main text).
>
> **Response 4: Single view in Table 2 (corresponding to quesiton #5)**
>
> - The “Single View” column refers to the results where only one sub-view of the image is provided to the model (only the initial view is available). This functions as a reference for the shifting evaluation when only insufficient perceptual field is available. In Table 2, the “Single View” refers to the initial view of shifting-R.
>
> Response 5-8 correspond to weaknesses concerning our pipelines:
>
> **Response 5: It is not clear why the mixed pipeline requires the model to discard images. (corresponding to weakness #1)**
>
> - There could be some misunderstandings. It is **not a required operation to discard images**. The case in Fig 4 (c) is just an example demonstrating that the model can discard an image if it determines that image to be irrelevant to the question after zooming. If it is useful, the image will  undoubtedly be retained. This approach is designed to simulate ideal active perception behaviors.

---

> > ### Author Response · Authors · 2024-11-25
> > **Response (Part 2)**
> >
> > **Response 6: All three pipelines are similar to each other (corresponding to weakness #1)**
> >
> >  - The three pipelines are largely distinguishable from each other. Each of the two fundamental settings, zooming and shifting, is designed to evaluate a distinct aspect of active perception under constrained views, where zooming restricts the resolution of views, and shifting restricts perceptual field of views. In contrast, the mixed setting does not focus on a single aspect. The mixed pipeline permits free-form operations without constraints compared to zooming and shifting. It simulates ideal active perception behavior by enabling models to autonomously zoom and/or shift and perform further actions (zoom and/or shift) as needed. **None of the operations are mandatory in mixed evaluation**. Models can freely decide to zoom, shift, or discard a previous selection. Furthermore, the mixed pipeline is reversible by allowing models to “discard” views after zooming/shifting into them if the model decides those views as unnecessary. However for the two fundamental settings (zooming or shifting) enforce their respective operations as mandatory and irreversible.
> >
> >  - Here is a detailed comparison of the pipelines:
> >
> >      - Zooming Pipeline: This pipeline addresses one of the basic ability of active perception. It focuses on **fine-grained information extraction** by allowing models to start with the global image view and selectively zoom in on subregions. It assesses the model’s ability to identify and process relevant details according to a given context.
> >      - Shifting Pipeline: This pipeline addresses the other basic ability of active perception. The shifting pipeline introduces an additional layer of challenge by restricting the initial global view. Models must actively explore the image, shifting their perceptual fields **to gather the missing information**. This pipeline emphasizes the model’s ability to navigate visual space incrementally, mimicking real-world scenarios where full context is unavailable.
> >      - Mixed Pipeline: The mixed pipeline combines both zooming and shifting mechanisms to simulate ideal active perception behavior. The key contribution lies in requiring models to **autonomously choose among operations to simulate ideal active perception behavior**. The decision to discard irrelevant images or focus on specific views adds a layer of complexity, as it pushes the model to prioritize relevant visual fields dynamically. This mimics real-world decision-making where both global and localized exploration are necessary.
> >
> > **Response 7: The shifting pipeline does not give the model agency over which views it wants to request ... without any penalty of requesting more views. (corresponding to weakness #2)**
> >
> > - We intentionally do not give the model agency over which views it wants to request in shifting pipeline to simulate human behavior when shifting perceptual fields for missing information. In real life, when humans look for more visual information by shifting the perceptual fields, the previously perceived views cannot simply be erased from memory. The shifting pipeline aims to examine if mdoels can realise the insufficiency of current visual information and decide to explore elsewhere for additional clues. This can be effectively satisfied via simple “next” and “stop” decisions.
> >
> > - The concern regarding “deciding which views to request” **is more relevant to the mixed evaluation pipeline**, which we have already implemented. It is beyond the scope of pure shifting operation to flexibly “deciding which views to request”. In contrast, in the mixed evaluation, models can flexibly choose to shift to certain views then discard them if model find them unnecessary after shifting, and then shift to other views as needed.
> >
> > - Regarding “the penalty of requesting more views”, we report the number of requested views of shifting setting in Table 4 as an indicator of the active perception ability of models. As noted in our paper, some models request all 4 views and still underperforms, indicating their limited active perception ability regarding view-shifting. We do not include penalties in this evaluation pipeline because the primary goal of our benchmark is **to fairly evaluate the active perception ability of models, not to improve it via mechanisms such as penalizing undesired behaviors**.

---

> > > ### Author Response · Authors · 2024-11-25
> > > **Response (Part 3)**
> > >
> > > **Response 8: A clarification of the shifting evaluation (corresponding to weakness #5, #6)**
> > >
> > > - We first evaluate the active perception abilities of MLLMs from two fundamental perspectives, zooming and shifting. Neither perspective **alone** is sufficient for a complete assessment of active perception. We are concerned about a possible misunderstanding that a single shifting pipeline alone is adequate for fully evaluating active perception. Instead, the shifting setting represents one critical aspect of active perception, measuring if the model can actively explore the image by shifting their perceptual fields **to gather the missing information**, and understanding **when to request additional views and when to stop to provide an answer**[1].
> > >
> > > Response 9-12 correspond to experimental results:
> > >
> > > **Response 9: Experimental results in Table 2 (corresponding to weakness #3)**
> > >
> > > - **Zooming vs Full**:
> > >     - We emphasize that the “Full Image” setting is not part of the evaluation for active perception ability. It rather serves as a reference only. Our focus is on evaluating the active perception abilities of MLLMs (e.g., zooming and/or shifting), rather than directly assessing question-answering accuracy.
> > >     - Final QA accuracy is not solely determined by correctly detecting the desired views. While accurate view selection often contributes to correct answers, as shown in the results of GPT-4o, Qwen2-VL, LLaVA-OneVision, and etc. For models such as Claude and Gemini, the Zooming accuracy decreases compared to the Full setting. We argue that splitting the information into separate views challenges the model’s ability to integrate and comprehend the data effectively. This reflects the limitations of some current models in selectively focusing on relevant subregions.
> > >
> > > - **Single View**:
> > >     - For the shifting evaluation, the “single view” serves as a reference, and is not a part of the evaluation of active perception ability.
> > >     - All the propreitary models exhibit good active perception ability given restricted range of views and are able to derive a combined understanding of all the split views. The cases where some models perform better on a single view than on shifting settings indicate that they cannot effectively integrate information across multiple views under constrained perceptual fields.
> > >
> > > - We've rearranged Table 2 to highlight that Full is not a part of Zooming evaluation, neither is Single View to Shifting evaluation.
> > >
> > > - **The necessity of active perception in ActiView**:
> > >     - While some images-question pairs may seem solvable without active perception, the benchmark is specifically designed, from the perspectives of zooming and shifting, to test whether models can:
> > >         - Correctly detect the regions containing key information to the query. (zooming)
> > >         - Recognize when additional information is needed. (shifting)
> > >         - Effectively gather and integrate that information under constrained perceptual settings. (zooming and shifting)
> > >     - The results observed in Table 2 suggest that many models are not yet capable of fully exhibiting these behaviors, validating the benchmark’s role in identifying and addressing these gaps.
> > >
> > > **Response 10: Analysis of experimental results on view selection (corresponding to weakness #4)**
> > >
> > > - **The concern on “Recall does not have a significant impact on the performance”**: We also provide the presicion, f1 and accuracy of zooming view selection here:
> > >
> > >   |  Model  | Recall |Presicion|F1 score|ACC(View Select)|ACC(Zooming QA)|
> > >   |:--------|:------:|:-------:|:---:|:--------------:|:-------------:|
> > >   | Claude | 67.64 |**81.49**|**73.92**|**68.46**|**72.31**|
> > >   | GPT-4o |**69.03**| 79.60 | 73.94 | 67.85 | 68.62  |
> > >   |Qwen2-VL | 64.61 |  72.55 |68.35| 60.46 | 64.62  |
> > >   |MiniCPM-V 2.6|57.03| 69.13  |62.50| 54.92 | 61.85  |
> > >   |mPLUG-Owl3| 68.57 |  68.25  |68.41| 58.15 | 60.92   |
> > >   | Idefics3|  41.09 |  71.31  |52.14| 50.15 | 58.15   |
> > >   | InternVL2 | 41.09 | 71.03 | 52.06 |50.00 | 56.00 |
> > >
> > >   - The recall, presicion, F1 score, and accuracy collectively indicate the overall trend of final question-answering accuracy in the zooming evaluation. The influence of view selection should be measured through all the above metrics, especially recall, F1 score and ACC (View Select). However, outliers may occur because the QA performance is also influenced by external factors such as the training corpus.

---

> > > > ### Author Response · Authors · 2024-11-25
> > > > **Response (Part 4)**
> > > >
> > > > **Response 11: Ablation where all 4 views is given to understand the distraction issue stated in L. 460 (corresponding to quesiton #6)**
> > > >
> > > > - For this ablation study, we directly provide all 4 views and require models (three top open-source models in Table 2, Qwen2-VL, Idefics3, and MiniCPM-V 2.6) to answer. When simultaneously providing all views, the accuracy of Qwen2-VL is 59.69%, and Idefics3 is 52.31%, compared to their respective Shifting-R results of 61.23% and 61.85%. **This suggests that such setting do distract the final reasoning of some strong models.**
> > > >
> > > > - However, for this experiment, MiniCPM-V 2.6 achieves 60.92%, higher than 54.77% of Shifting-R. When iteratively shifting views, both correct and distracting information are reinforced, and the process usually stops once the model recognises the information as sufficient. Repetition of information is not the only factor affecting the final results. Other factors unrelated to our addressed active perception, such as the image order and the model structure, also influence to some extent.
> > > >
> > > > - Case (b) in Fig. 5 is an example for this issue. Although the model is finally presented with the correct information, the previously presented distracting information (e.g. the 69 of papaya, which is highest price among all presented fruits) is also reinforced throughout the process.
> > > >
> > > > **Response 12: Misunderstanding of showcases (corresponding to the last weakness)**
> > > >
> > > > - It seems that each step taken by the model from the perspective of active perception is correct in this case. The correct information appears at the last shifted view, but all the other views are presented to the model as well. In real life, when humans look for more visual information by shifting the perceptual fields, the previously perceived views cannot simply be erased from memory. Similarly, if any misleading information exists, it cannot be removed delibarately (by us) from the model input. **Selecting the correct view only demonstrates that the model has chosen the correct location.** However, an incorrect answer to the question indicates that the model has not effectively addressed other inherent aspects of active perception, such as “understanding the reasons for sensing, choosing what to perceive, and determining the methods, timing, and locations for achieving that perception.” [1] For example, the model may have failed to accurately perceive that the price of 69 corresponds to papaya, not watermelon.
> > > >
> > > > [1] Bajcsy, R., Aloimonos, Y., & Tsotsos, J. K. (2018). Revisiting active perception. Autonomous Robots, 42, 177-196.

---

> ### Author Response · Authors · 2024-11-27
>
> Dear reviewer,
>
> We hope our responses have addressed your concerns and look forward to further discussions.

---

> > ### Comment · Reviewer_5Vjd · 2024-11-29
> >
> > I would like to thank the authors for their response to my initial review.
> >
> > While some of my questions could be addressed, key concerns from my review remain. These include:
> >
> > - While I appreciate the elaboration of the different pipelines, my opinion still holds that especially zooming and mixed are very similar, and it is not clear how important the ability to discard images really is. As a result, the claimed contribution of introducing a mixed pipeline is not convincing.
> > - For the proposed benchmark to be effectively evaluating active perception, there needs to be a performance difference between using active perception vs. baseline inputs to the model. For instance, if the global image is more effective for solving the task, it indicates that the benchmark is not difficult enough to require finding fine-grained information not visible in the global image alone. This could be validated by providing the best local views to the model (e.g. containing all human-annotated clues) to confirm that performance can increase if active perception is successful. A useful benchmark would maximize the discrepancy between these two baselines such that improving active perception abilities of a model can close this gap.
> >   - The currently provided baselines of "Full Image", "Single Views", and "All Views" together with the results on the difficulty settings for shifting indicate that the benchmark does not challenge a model to be good at active perception, i.e., it does not primarily evaluate active perception ability, but (multi-)image comprehension more generally.
> >   - While it is understandable that humans use active perception, from this study, it is not clear that MLLM require it. Due to the small gap between using active perception or simply providing the global image, MLLMs might be better off not performing active perception. Instead providing the global image is often just as good. This is simply speaking from a practical point of view. A new benchmark should present a setting that makes active perception necessary.
> >
> > For these reasons, I believe the paper requires a more substantial revision to improve the value of the benchmark for the research community. I hope this feedback helps in achieving that. As a result, I keep my initial score.
> >
> > Other comments/suggestions:
> > - Response 1: It seems a bit inflated to count the questions 5x when there is not too much differences, e.g., in the difficulties of the shifting task.
> > - Response 3: The random performance can be calculated exactly without the need of 10k random answer simulations.
> > - Response 11: The example you give in Case (b) in Fig. 5 is exactly an example where being good at active perception makes little difference if the model cannot actively choose to receive distracting information or not. It relates to not having enough agency in this setting such that the benchmark effectively evaluates other aspects of the model (ignoring distractions, processing multiple images, reasoning with a lot of visual information) instead of evaluating active perception.

---

### Official Review · Reviewer_sZTj · 2024-11-04

**Soundness:** 3
**Presentation:** 3
**Contribution:** 3
**Rating:** 6
**Confidence:** 4

**Summary:**

The paper introduces ActiView, a benchmark designed to evaluate the active perception abilities of multimodal large language models (MLLMs) through a visual question answering (VQA) format. Active perception involves comprehending the reasons for sensing, selecting what to perceive, and deciding on the methods, timing, and locations needed to achieve that perception. The study, which tested 27 models, found that the ability to handle multiple images simultaneously is vital for effective active perception. While all models outperformed random guessing, they still fell short of human performance. The authors highlight variations in performance among model types and the capacity of single-image models to leverage multi-view information. ActiView establishes a framework for assessing and enhancing MLLM active perception, underscoring its significance for future research.

**Strengths:**

1. This paper explores the active perception capabilities of MLLMs and introduces a novel benchmark “ActiView”.
2. The authors present three distinct evaluation pipelines—Zooming, Shifting, and Mixed—specifically designed to assess the models' active perception abilities.
3. The paper includes comprehensive experiments and analyses that yield insightful conclusions.

**Weaknesses:**

1. Evaluation Pipelines: The current evaluation pipelines rely on fixed image splitting strategy, resulting in notable discrepancies between the evaluation pipeline and the ideal active perception behaviors. For example, in the Zooming pipeline, the model can not perform additional zooming on the selected sub-image. Similarly, in the Shifting pipeline, the viewpoint movement is constrained to a predetermined granularity.

2. Insufficient Detail Analysis: As observed in Table 2, several models, such as Gemini-1.5-pro and Claude 3.5 Connect, exhibit superior performance on ‘full images’ compared to the ‘zooming’. A more detailed analysis of the underlying reasons for this discrepancy is warranted.

3. According to the prompts provided in the appendix I, it appears that the Mixed pipeline accounts for direct response scenarios (allowing models to specify the number of parts to zoom in or respond with "none" if no zooming is needed). However, the Zooming pipeline does not allow for a 'none' response. An explanation for this inconsistency would be beneficial.

**Questions:**

See the weknesses section for more details.

---

> ### Author Response · Authors · 2024-11-24
> **Response to weaknesses**
>
> Thanks for your comments, please find our responses as follows:
>
> **Response 1: Image splitting strategy for evaluation pipelines (corresponding to weakness #1).**
>
> - We aim to address the significance of active perception ability, and introduce a set of splitting strategy for the two fundamental abilities of active perception, i.e., zooming and shifting. Additional experiments on different splitting strategies for both zooming and shifting are provided in Appendix F1. Results suggest that different splitting strategies inconsistently affect the performance, presenting a trade-off between zooming and shifting performances along with the increasing of splits. We propose **using the 4-split setting to balance zooming and shifting evaluations**. This fixed image splitting strategy already demonstrates discrepancies in active perception ability between models and humans, and among different models as well.
>
> - In addition to the basic settings for zooming and shifting, we also propose a more dynamic and natural setting, **the mixed evaluation, to simulate the ideal active perception behavior**. This setting incorporates both zooming and shifting without restrictions on how should models behave. Models can autonomously zoom and/or shift, and perform further actions (zoom and/or shift) if deemed necessary. This **can be easily adapted to finer granularity evaluation of either zooming or shifting** by limiting the type of behavior and combining it with different splitting strategies as in Appendix F1 (such as symmetric splitting, e.g. 2 $\times$ 2 grids, and 3 $\times$ 3 grids, and asymmetric splitting, e.g. 2 $\times$ 3, 2 $\times$ 4).
>
> **Response 2: Detailed analysis compared to “Full” image (corresponding to weakness #2)**
>
>  - First, we must clarify that the “Full Image” setting is not part of the evaluation for active perception ability. It is provided for reference only. Our primary focus is on evaluating the active perception abilities of MLLMs (e.g., zooming and/or shifting), rather than directly measuring question-answering accuracy.
>
>  - Models like Gemini-1.5-pro and Claude 3.5 Connect may perform better with “full images” because they can access the complete context, allowing them to process visual information holistically. In contrast, the zooming pipeline introduces constraints on the perceptual fields by requiring models to focus on split views. This challenges their ability to integrate multiple dependent details into broader understandings. And the performance drop of models (compared to the full setting) implies the certain weaknesses in active perception ability for models. The weakness could be (but not restricted to) the ability to integrate multiple dependent details, or recognise key information regarding a given question. This discrepancy highlights the importance of the zooming pipeline in exposing weaknesses in a model’s active perception abilities.
>
> **Response 3: Zooming pipeline does not allow for a “none” response. (corresponding to weakness #3)**
>
>  - We did not strictly specify the number of selected views for the zooming pipeline, meaning a “none” selection is permitted. While we recommend selecting at least one view, this is not a mandatory requirement. Some models respond with empty strings (“”), provide a reason for not selecting a view, or explicitly output “none” when required to output the image splits for zooming. We regard these responses as “none” for the zooming selection, and only the full images are provided instead of zoomed splits in this case.
>
>  - For example, 1% of Brote-IM-XL’s responses were “none”, and so were 4.9% of Mantis’s responses.

---

> > ### Comment · Reviewer_sZTj · 2024-11-30
> >
> > Thanks for the detailed responses to my review comments. I appreciate the clarifications provided, particularly on the image splitting strategy and the flexibility of the zooming pipeline. I am inclined to maintain a positive evaluation.

---

### Official Review · Reviewer_XwV7 · 2024-11-04

**Soundness:** 3
**Presentation:** 3
**Contribution:** 2
**Rating:** 5
**Confidence:** 5

**Summary:**

This paper takes an "Active Perception" perspective for multimodal LLMs and proposes an evaluation benchmark where VQA is chosen as the proxy task.  Given an image, the task is to actively zoom in or out or shift perceptual field based on reasoning to answer the questions successfully.  Experiments are performed on several models and the results reveal a performance gap.

**Strengths:**

1. Active perception / active vision has been a long standing research field but has been sidelined by static dataset-based training pipelines (for not very convincing reasons). Active perception is rooted in neuroscience, cognitive psychology, and biology.  This paper brings a fresh perspective.
2. The paper is well written -- figures and tables help the flow.
3. Several models are tested and experimental settings are sound and transparent.

**Weaknesses:**

1. For both zooming or shifting, the image resolution/ratio will either remain the same (for example, zooming from a 400x300 image into a 80x60 patch) or will be resized (as stated in line 330 onwards). But sometimes, segmentation might be necessary instead of looking at the entire w x h patch.  This has not been explored.  While I understand that the benchmark only contains zoom/shift for now, what are some other operations that could be integrated into the benchmark?
2. The benchmark could be made a lot more comprehensive in terms of size: there are 1625 evaluation instances which means that approximately getting 16 questions wrong leads to a 1% performance drop.  In Table 1, the drop in performance compared to Full Image doesn't seem very statistically significant if it's just a few percentage points.
3. The benchmark could be made a lot more comprehensive in terms of tasks: why just VQA? Why not extend the annotations to other tasks such as captioning, vision-language inference (see NLVR2, or SNLI-VE etc.), image-text retrieval, etc.

**Questions:**

1. This is an ICLR submission, but to me this could be a CVPR/ICCV/ECCV paper as it is a "pure vision" paper.  I'd like the authors to provide more insights into what value the work could bring to representation learning and if the work has broader impact beyond vision.  This is **not** a criticism of the paper (which is why I haven't listed it above as a weakness), but I'd like to engage into a conversation about this.

---

> ### Author Response · Authors · 2024-11-24
> **Response to weaknesses**
>
> Thanks for your comments, please find our responses as follows:
>
> **Response 1: Discussion over segmentation and other operations (corresponding to weakness #1).**
>
>  - Intuitively, segmentation is a variant of our zooming setting, allowing models to see on object segments or irregular regions in an image. Although segmentation ensures semantic integrity (compared to regular patches), our approach allows multiple patches to be selected to maintain the completeness of semantic segments, even when they are divided into different patches.
>
>  - We did not include this specific setting for the following reasons:
>
>     - Most current models do not support directly reading segments. Some models claim to allow segment inputs by converting them into w $\times$ h (according to the segment size) patches for processing. Our pipeline of splitting the whole image into smaller views already presents significant challenges for these models.
>     - While this setting is natural and reasonable for evaluation, our experiments showed that current models often fall short to follow instructions (to output correct segments) or suffer from significant hallucinations (producing incorrect segments). These issues result in near-random performance, even for advanced models like GPT-4.
>     - As a compromise, we proposed the zooming setting and conducted experiments with varying numbers of image splits. The results are presented in Appendix F1.
>
>  - With improvements in model capabilities, we hope future models will be able to satisfy these requirements. We have been working on and continue to explore this setting both for evaluation purposes and to improve model performance.
>
> - For now we only consider zooming and/or shifting for MLLMs. In the future, fine-grained grounding operations such as iteratively responsing with bounding boxes for the deducted visual clues could be considered. Also, the detection of visual redundency (information do not help with answering) and distractive information (information that doom to failure) could also be considered. Regarding the active perception ability, more modalities should be included beyond vision in the future, and operations such as modality division and key modality determination are also worth investigating.
>
> - For now, we focus on zooming and/or shifting for multimodal language models (MLLMs). In the future, fine-grained grounding operations, such as iteratively responding with bounding boxes for deduced visual clues, could be explored. Also, detecting visual redundancy (information that does not contribute to answers) and distractive information (misleading information that doom to failure) could be considered. Additionally, for active perception abilities, incorporating other modalities beyond vision is a future trend for research and practical applications. In this situation, operations such as modality division and key modality determination, will be worth investigating.
>
> **Response 2: The significance of results (corresponding to weakness #2)**
>
> - We emphasize that the “Full Image” setting is not part of the evaluation for active perception ability. It rather serves as a reference only. Our focus is on evaluating the active perception abilities of MLLMs (e.g., zooming and/or shifting), rather than directly assessing question-answering accuracy.
>
> - Most current models are fine-tuned for VQA tasks, giving them an inherent ability to “guess” answers. Some models guess the correct answer without truly understanding the underlying reasons or clues. We view our proposed zooming, shifting, and mixed settings as OOD tasks compared to traditional image-question pair evaluations that simply gives an image-question pair and requires models to answer. Considering this feature, even small differences (e.g., 1%) in performance are significant within the context of our benchmark. A few percentage points can represent a consistent inability to reason under our constrained settings.
>
>  **Response 3: Why just VQA? (corresponding to weakness #3)**
>
>  - VQA is a suitable format for evaluating various visual-language tasks. For most current MLLMs, tasks such as NLVR2, SNLI, and image retrieval can be adapted to the QA format via prompt design.
>
> - We adopt the VQA format because it provides a easy-to-understand, clear and focused evaluation of active perception abilities. VQA inherently requires models to reason over visual information, making it an ideal task for assessing capabilities including zooming and shifting. By concentrating on this task, our benchmark effectively highlights and evaluates active perception without blurring the focus.

---

> > ### Author Response · Authors · 2024-11-24
> > **Response to questions**
> >
> > **Response 4: Regarding the submission track (corresponding to question)**
> >
> >  - We do not consider this paper as a “pure vision” paper appropriate only for CVPR/ICCV/ECCV. It focus on the active perception ability that requires not only visual recognition and reasoning, but also desicion-making, planning, and etc. These are critical skills not only for “vision”, but also for understanding and interpreting of multimodal representations, reasoning behavior of models, structure design, and etc. Our benchmark aims at evaluating the active perception ability, which is applicable not just to “pure vision” topics, but to embodied, robotics and agent-based research as well.
> >
> >  - Moreover, ICLR is an inclusive conference that welcomes a broad range of topics in machine learning, including vision, audio, speech, language, music, robotics, healthcare, and more (as stated in the official ICLR announcement). ICLR explicitly supports vision-related tracks in the official website, such as “representation learning for computer vision, audio, language, and other modalities”, and includes a track for “datasets and benchmarks.”
> >
> >  - In conclusion, we believe our paper is appropriately submitted to the right conference.

---

> ### Author Response · Authors · 2024-11-27
>
> Dear reviewer,
>
> We hope our responses have addressed your concerns and look forward to further discussions.

---

> > ### Comment · Reviewer_XwV7 · 2024-11-27
> >
> > I appreciate the authors' rebuttal. However, the experimental scope of this paper remains a significant concern. The authors have rightly identified several areas for expansion/improvement/further investigation in their rebuttal -- I encourage the authors to pursue these directions and I'm confident that it will add value to future versions of the paper.
> >
> > At the moment, I do not believe the weaknesses that I mentioned have been sufficiently addressed so I will stay will my original rating.

---

### Note · Authors · 2024-12-15

I have read and agree with the venue's withdrawal policy on behalf of myself and my co-authors.